# The E2 glycoprotein holds key residues for Mayaro virus adaptation to the urban *Aedes aegypti* mosquito

**Chelsea Cereghino**[1,2©], **Ferdinand Roesch**[3,4©], **Lucía Carrau**[3,5], **Alexandra Hardy**[3], **Helder V. Ribeiro-Filho**[6], **Annabelle Henrion-Lacritick**[7], **Cassandra Koh**[7], **Jeffrey M. Marano**[1,8], **Tyler A. Bates**[1], **Pallavi Rai**[1], **Christina Chuong**[1], **Shamima Akter**[1,9], **Thomas Vallet**[3], **Hervé Blanc**[7], **Truitt J. Elliott**[10,11,12], **Anne M. Brown**[10], **Pawel Michalak**[1,13,14,15], **Tanya LeRoith**[1], **Jesse D. Bloom**[16,17], **Rafael Elias Marques**[6], **Maria-Carla Saleh**[7], **Marco Vignuzzi**[3], **James Weger-Lucarelli**[1,2,3]*

1 Department of Biomedical Sciences and Pathobiology, VA-MD Regional College of Veterinary Medicine, Virginia Tech, Blacksburg, Virginia, United States of America, 2 Center for Emerging, Zoonotic, and Arthropod-borne Pathogens, Virginia Tech, Blacksburg, Virginia, United States of America, 3 Institut Pasteur, Viral Populations and Pathogenesis Unit, Centre National de la Recherche Scientifique UMR 3569, Paris, France, 4 UMR 1282 ISP, INRAE Centre Val de Loire, Nouzilly, France, 5 Department of Microbiology, New York University Langone Medical Center, New York, New York, United States of America, 6 Brazilian Biosciences National Laboratory, Brazilian Center for Research in Energy and Materials (CNPEM), Campinas, São Paulo, Brazil, 7 Institut Pasteur, Viruses and RNA Interference Unit, Centre National de la Recherche Scientifique UMR 3569, Paris, France, 8 Translational Biology, Medicine, and Health Graduate Program, Virginia Tech, Roanoke, Virginia, United States of America, 9 Department of Bioinformatics and Computational Biology, School of Systems Biology, George Mason University, Fairfax, Virginia, United States of America, 10 Program in Genetics, Bioinformatics, and Computational Biology (GBCB), Virginia Tech, Blacksburg, Virginia, United States of America, 11 Research and Informatics, University Libraries, Virginia Tech, Blacksburg, Virginia, United States of America, 12 Department of Biochemistry, Virginia Tech, Blacksburg, Virginia, United States of America, 13 Edward Via College of Osteopathic Medicine, Monroe, Louisiana, United States of America, 14 Center for One Health Research, VA-MD Regional College of Veterinary Medicine, Blacksburg, Virginia, Untied States of Ameria, 15 Institute of Evolution, University of Haifa, Haifa, Israel, 16 Basic Sciences Division and Computational Biology Program, Fred Hutchinson Cancer Research Center, Seattle, Washington, United States of America, 17 Howard Hughes Medical Institute, Chevy Chase, Maryland, United States of America

© These authors contributed equally to this work.

* weger@vt.edu

**Data Availability Statement:** Raw sequencing data was deposited under the Sequence Read Archive (SRA, NCBI) under bioproject number

## Abstract

Adaptation to mosquito vectors suited for transmission in urban settings is a major driver in the emergence of arboviruses. To better anticipate future emergence events, it is crucial to assess their potential to adapt to new vector hosts. In this work, we used two different experimental evolution approaches to study the adaptation process of an emerging alphavirus, Mayaro virus (MAYV), to *Ae. aegypti*, an urban mosquito vector of many other arboviruses. We identified E2-T179N as a key mutation increasing MAYV replication in insect cells and enhancing transmission after escaping the midgut of live *Ae. aegypti*. In contrast, this mutation decreased viral replication and binding in human fibroblasts, a primary cellular target of MAYV in humans. We also showed that MAYV E2-T179N generates reduced viremia and displays less severe tissue pathology *in vivo* in a mouse model. We found evidence in mouse fibroblasts that MAYV E2-T179N is less dependent on the Mxra8 receptor for replication than WT MAYV. Similarly, exogenous expression of human apolipoprotein receptor 2

PRJNA796940. Other relevant data are within the manuscript and its Supporting Information files.

**Funding:** This work was funded by the DARPA PREEMPT program administered through DARPA Cooperative Agreement HR001118S0017, this funding was awarded to M-C.S., M.V, and J.W-L. This work also received funding from Laboratoire d'Excellence Integrative Biology of Emerging Infectious Diseases (grant ANR-10-LABX-62-IBEID) to M-C.S. and M.V. Further support was provided by startup funds awarded to J.W-L by the Virginia-Maryland College of Veterinary Medicine and a grant from the One Health Research Funding Program awarded to J.W-L and P.M. The funders had no role in study design, data collection and analysis, decision to publish, or preparation of the manuscript.

**Competing interests:** The authors have declared that no competing interests exist.

and Mxra8 enhanced WT MAYV replication compared to MAYV E2-T179N. When this mutation was introduced in the closely related chikungunya virus, which has caused major outbreaks globally in the past two decades, we observed increased replication in both human and insect cells, suggesting E2 position 179 is an important determinant of alphavirus host-adaptation, although in a virus-specific manner. Collectively, these results indicate that adaptation at the T179 residue in MAYV E2 may result in increased vector competence–but coming at the cost of optimal replication in humans–and may represent a first step towards a future emergence event.

## Author summary

Mosquito-borne viruses must replicate in both mosquito and vertebrate hosts to be maintained in nature successfully. When viruses that are typically transmitted by forest dwelling mosquitoes enter urban environments due to deforestation or travel, they must adapt to urban mosquito vectors to transmit effectively. For mosquito-borne viruses, the need to also replicate in a vertebrate host like humans constrains this adaptation process. Towards understanding how the emerging alphavirus, Mayaro virus, might adapt to transmission by the urban mosquito vector, *Ae. aegypti*, we used natural evolution approaches to identify several viral mutations that impacted replication in both mosquito and vertebrate hosts. We show that a single mutation in the receptor binding domain of E2 increased transmission by *Ae. aegypti* after bypassing the midgut infection barrier but simultaneously reduced replication and pathology in a mouse model. Mechanistic studies suggested that this mutation decreases the dependence of MAYV on human Mxra8 and the putative MAYV receptor human ApoER2 during replication. This suggests MAYV with this mutation alone is unlikely to be maintained in a natural transmission cycle between mosquitoes and humans. Understanding the adaptive potential of emerging viruses is critical to preventing future pandemics.

## Introduction

Arthropod-borne viruses (arboviruses) impose significant economic and public health costs worldwide. Zika virus (ZIKV), dengue virus (DENV), chikungunya virus (CHIKV) and West Nile virus (WNV) are now globally established and have caused significant outbreaks. In past decades, they mainly affected developing countries in tropical regions of the world and have often been neglected. Recently, local cases of ZIKV, DENV and CHIKV [1,2] have been detected in Europe and the United States, demonstrating that more temperate areas of the world are endangered by arboviruses, and that preventing arboviral emergence is one of the key public health challenges for years to come.

Mayaro virus (MAYV; Genus *Alphavirus*) was first isolated in Trinidad and Tobago in 1954 where it was associated with cases of mild febrile illness [3]. Since then, MAYV has caused sporadic outbreaks in several countries in South and Central America, including Brazil [4], Mexico [5], Peru [6], French Guiana [7], Bolivia [8], Ecuador [9] and Venezuela [10]. Imported cases have been described in multiple European countries including the Netherlands [11], France [12], Germany [13] and Switzerland [6]. Three genotypes have been described for MAYV: D (widely dispersed), L (limited) and N (new) [14]. However, recombinant strains of MAYV may also circulate in the Amazon basin [15]. Despite the increase in epidemiological

data and surveillance in recent years, MAYV circulation is still likely under-estimated. Many factors contribute to this underreporting, including antibody cross-reactivity which complicates serological studies [6], significant overlap in the areas of distribution of MAYV and CHIKV, and the similarities between the clinical manifestations caused by MAYV and other arboviruses [16]. Like CHIKV and DENV, MAYV infection induces a febrile illness, with symptoms such as fever, rash, headaches, and nausea [17]. MAYV also induces joint pain in most symptomatic patients [18], which sometimes lasts for several months [19] and may be caused by sustained production of pro-inflammatory cytokines [20,21]. In rare cases, Mayaro fever is associated with more serious clinical outcomes, such as neurological complications, hemorrhagic manifestations or death [22]. Antibodies directed against MAYV have been observed in many animal species beyond humans, including primates, birds, and rodents [4,23], which may act as reservoir species.

MAYV is considered a serious candidate for viral emergence [14,24,25]. While most MAYV outbreaks so far have happened in rural areas in close proximity of tropical regions [26] or in indigenous communities [27], there is increasing concern that MAYV may escape this sylvatic cycle and become urbanized. Indeed, MAYV circulation has recently been observed in urban areas of Haiti [15]. In such areas, efficient transmission by *Ae. aegypti* would likely increase the risk of a sustained outbreak. Thus, the current epidemiological situation of MAYV is reminiscent of that of CHIKV before it emerged in 2006, when a single mutation in E1 allowed for better adaptation to *Ae. albopictus*, ultimately leading to the CHIKV outbreak in the Indian Ocean region [28].

While it is well established that MAYV can infect forest-dwelling *Haemagogus* mosquitoes in the context of its sylvatic cycle of MAYV [29], it is unclear whether MAYV can be transmitted in nature by mosquito vectors suited for urban transmission. Several laboratory studies have suggested that *Ae. aegypti* [30–32] and *Ae. albopictus* [33,34] mosquitoes can be readily infected by MAYV and may support viral transmission, albeit to different levels. One report also suggested that *Anopheles* mosquitoes may also participate in MAYV transmission [35]. While these studies suggest that MAYV has the potential to be transmitted by urban vectors, definitive evidence that this happens outside of laboratory settings is still lacking.

In this work, we sought to test the hypothesis that similar to CHIKV pre-emergence, MAYV is not yet fully adapted to *Ae. aegypti* transmission. We used two different *in vitro* evolution approaches to identify the genetic blocks that MAYV must overcome to better adapt to the anthropophilic vector *Ae. aegypti*. Deep mutational scanning and serial passaging approaches both identified T179N as a key mutation in the receptor binding protein, E2, resulting in increased MAYV transmission by *Ae. aegypti* after overcoming an initial infection barrier. This mutation decreased viral replication in human and mouse cells and in mice, and during replication was associated with decreased dependence on human Mxra8 and a putative human MAYV receptor apolipoprotein E receptor 2 (ApoER2), suggesting that adaptation of MAYV to this new vector may come at an evolutionary cost. This may explain why this mutation has not arisen yet in nature. Evolution at this residue may constitute an important first step towards increasing MAYV transmission in *Ae. aegypti*, although other barriers to infection of the mosquito midgut and human cells must be overcome to result in sustained transmission.

## Results

### Natural evolution and deep mutational scanning enrich for similar high frequency mutations in MAYV

We sought to identify mutations enabling MAYV adaptation to mammalian and insect hosts using two complementary approaches: naturally evolving virus through serial passaging and

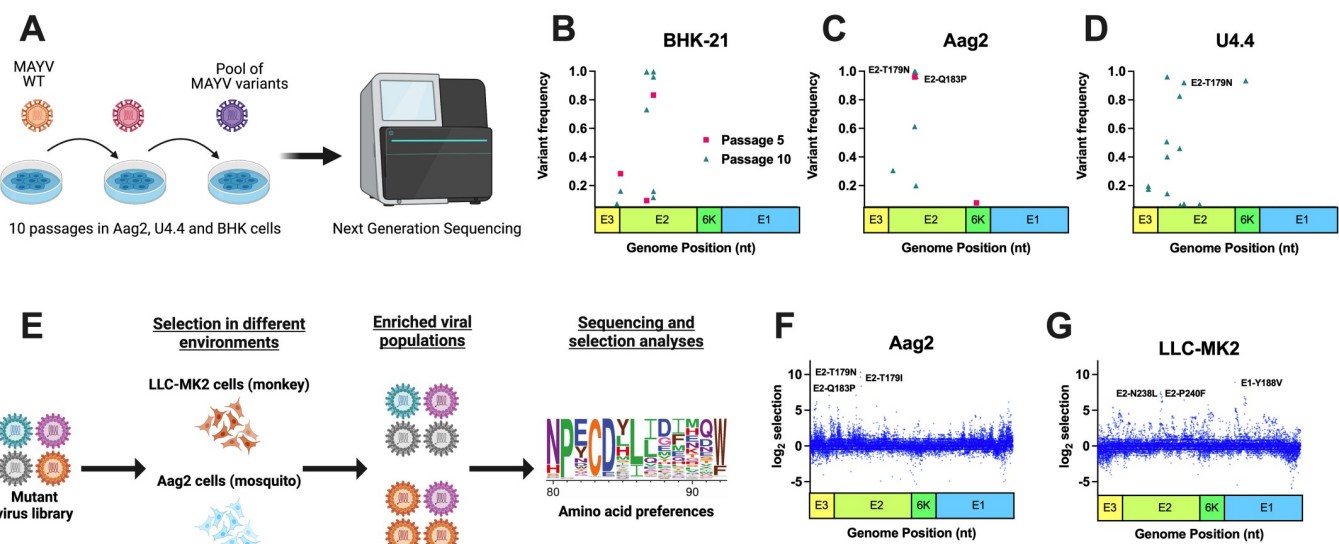

**Fig 1. Natural evolution and deep mutational scanning (DMS) enrich for similar high frequency mutations in Mayaro virus (MAYV).** Experimental evolution was performed using traditional serial passaging and DMS to identify molecular determinants of emergence. **A:** Traditional experimental evolution was performed by serially passaging MAYV at a MOI of 0.01 in BHK-21 cells (hamster) and two insect cell lines, Aag2 (*Ae. aegypti*) and U4.4 (*Ae. albopictus*). A total of ten passages were performed. **B-D:** Following one, five, and ten passages, the viral RNA was sequenced using Illumina NGS to identify potentially adaptive mutations. No high frequency variants were identified following passage one, and as such are not depicted here. **E:** The three MAYV DMS populations, along with WT MAYV, were used to perform three passages in Aag2 (**F**) and LLC-MK2 (**G**) cells. Following passage, the viral RNA was sequenced, and selection analyses were performed to identify enriched variants. The top three variants in each environment, based on selection strength, are presented for each graph. **A and E** were created with BioRender.com whose terms and conditions are at biorender.com/terms.

facilitating evolution through deep mutational scanning (DMS). Serial passaging was performed in BHK-21 (baby hamster kidney fibroblast), Aag2 (*Ae. aegypti* mosquito cells), U4.4 (*Ae. albopictus* mosquito cells), and 4a-3A (*Anopheles gambiae* mosquito cells) cells. Our rationale for using *Anopheles* cells was based on a recent report showing them to be competent for MAYV [35]. We sequenced the virus using Illumina next-generation sequencing (NGS) after passage 1, 5 and 10 (**Fig 1A**). All variants were below our threshold based on coverage depth and frequency in the passage 1 samples. As expected, most high frequency mutations were observed in the passage 10 samples, and the vast majority were found in the genes encoding for the envelope proteins, specifically E2 (**Fig 1B and 1D**). We only present mutations in the envelope proteins since few mutations in other protein-coding sequences were identified. Of note, we observed a consensus change in 3/6 replicates from Aag2 cells (with frequencies ranging from 0.58 to 0.97) at nucleotide position 8918 (**Fig 1C**), which resulted in a change within E2 at amino acid position 179 from a threonine (T) to either an isoleucine (I) or an asparagine (N). Mutations were also observed at this site for U4.4 cells (**Fig 1D:** 2/6; range of 0.07–0.89), but not mammalian cells (**Fig 1B**). Following passage in 4a-3A cells, we observed a consensus level mutation within E2 in 4/6 replicates at amino acid position 232 resulting in a change from histidine (H) to proline (P) (**S1 Fig**). While not the focus of these studies, future studies should study the impact of this mutation on adaptation potential for *Anopheles* mosquitoes. **S1 File** presents all the variants that passed our threshold, along with their frequencies and sequence coverage.

MAYV DMS populations (**Fig 1E**) were generated for all envelope genes (E3-E2-6K-E1; **S2A Fig**) in 3 independent populations, as previously described for other viruses [36–38]. DMS is a powerful approach to create a diverse viral population that theoretically contains all possible amino acid substitutions at each site within a given genomic region. Following selection in an environment, deep sequencing is performed, and computational analyses then

identify sites that are enriched or decreased. As expected, the initial DMS populations were highly diverse (**S2B Fig**) as assessed by nucleotide diversity in the DMS and WT MAYV populations. We then passaged the DMS viruses in LLC-MK2 (immunocompetent rhesus monkey kidney epithelial) and Aag2 cells at a MOI of 0.01 for three passages. Virus titers were relatively stable throughout passaging in LLC-MK2 (**S2C Fig**) and Aag2 (**S2D Fig**). Following sequencing, we observed several amino acid changes that were enriched in Aag2- and LLC-MK2-passaged DMS populations (**Fig 1F and 1G**). **S2** and **S3** **Files** present the full DMS selection outputs, including mutations enriched and selected against. Interestingly, the top three sites under selection (T179N, T179I, and Q183P) in Aag2 cells were in the E2 protein and were found as consensus changes in a least one of the replicates during natural evolution in Aag2 or U4.4 cells, though T179N and T179I were the most abundant in traditional passaging. The top sites under selection in LLC-MK2 were in E1 (Y118V) and E2 (P240F and N238L). We also passaged the DMS populations in live mice and *Ae. aegypti* and *Ae. albopictus* mosquitoes (**Fig S2E and S2F** and **S2 File**); however, we observed only weak signals of selection (**S3 Fig**). For the remaining studies, we will focus on mutations that were identified *in vitro*.

## Adaptive mutations to *Ae. aegypti* cells come at a fitness cost

We next generated several mutants identified in NGS data from both traditional passaging and DMS studies in order to measure the fitness of individual mutations in multiple environments (**Fig 2A**). The fitness of mutants was measured in a high-throughput fitness screen in several cell lines against a neutral, genetically marked competitor virus, as previously described [39–41]. Two of three mutations that were enriched during Aag2 passaging (E2-T179N, p<0.0001;

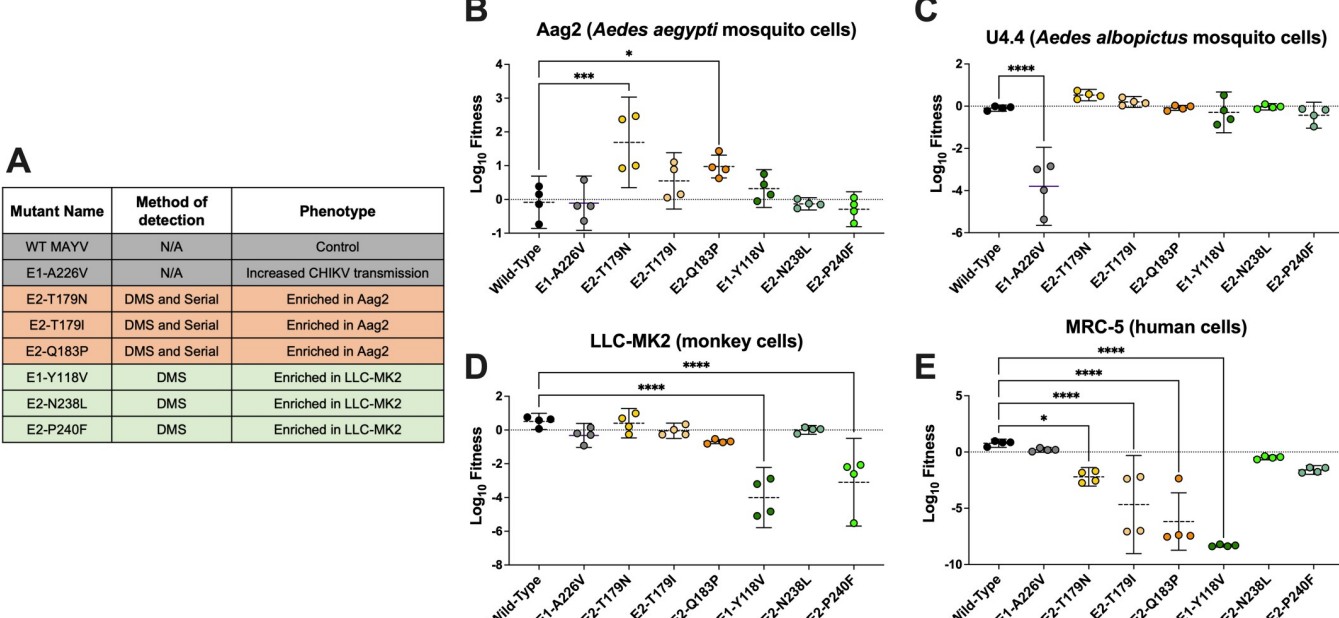

**Fig 2. Variants enriched in an *Ae. aegypti* cell line (Aag2) have increased fitness in Aag2 cells but decreased fitness in a human cell line (MRC-5). A:** List of the viruses used for the competition assays. The method (serial passaging or DMS) used to identify the mutations, as well as the phenotypes observed for these different viruses are indicated. **B-E:** Competition assays in mosquito (Aag2 and U4.4) and mammalian (LLC-MK2 and MRC-5) cell lines: Aag2 (**B**), U4.4 (**C**), LLC-MK2 (**D**), and MRC-5 (**E**) cells. Cells were infected at an MOI of 0.01 using a 1:1 ratio based on PFUs for each mutant or WT with a genetically marked MAYV competitor virus. Viral supernatants were harvested at 72h post-infection for Aag2 and 48h post-infection for the other cell lines. Replication of WT and mutant viruses was assessed by RT-qPCR using specific probes labeled with different fluorophores. Log$_{10}$ fitness was calculated by normalizing replication of each virus against a genetically marked reference virus. The mean of 4 independent experiments is represented with standard deviation. Statistical analysis: * = p<0.05; ** = p<0.01; *** = p<0.001; **** = p<0.0001 (one-way ANOVA with Dunnett's correction). "Serial" refers to the serial passaging natural evolution experiments to find adaptive mutations.

and E2-Q183P, p = 0.0036) significantly increased viral fitness in Aag2 cells in comparison to WT MAYV (**Fig 2B**). In U4.4 cells, the fitness of E2-T179N trended higher than WT, although this difference did not reach significance and is considered neutral along with mutation E2-Q183P (p = 0.0624; **Fig 2C**). These results suggest that MAYV E2-T179N and, to a lesser extent E2-Q183P, which were identified by two distinct passaging methods, demonstrate increased replication in insect cells but in a species-specific manner. In LLC-MK2 cells, two of the three mutations enriched in LLC-MK2-passaged DMS populations showed significant fitness losses (E1-Y118V, p<0.0001; E2-P240F, p<0.0001; **Fig 2D**). E1-Y118 also had lower fitness in MRC-5 human fibroblast cells (p<0.0001; **Fig 2E**). We also observed fitness losses for all Aag2-enriched variants in MRC-5 (p<0.001 for all comparisons to WT), which we did not observe in LLC-MK2. We then evaluated the fitness of mutations enriched following passage through live mice and mosquitoes and observed mostly modest effects on fitness in Aag2, U4.4, and LLC-MK2 cells (**S4 Fig**). In contrast, 50% of variants enriched in mice and 66% of variants enriched in mosquitoes demonstrated fitness losses in MRC-5 cells (**S4E Fig**). Notably, the E2-I354D mutant, which was enriched following passage in mice, lost fitness in both mammalian and mosquito environments.

To confirm the effect of the E2-T179N mutation on viral replication, we performed viral growth curves in Aag2, U4.4, and MRC-5 cells. Results were consistent with the competition assays, with a positive effect of E2-T179N on MAYV replication in Aag2 mosquito cells (p = <0.0001 at 1 d.p.i.) and U4.4 mosquito cells (p = 0.0262 at 1 d.p.i.), coming at the cost of reduced replication in MRC-5 human cells (p = 0.0021 at 1 d.p.i.) (**Fig 3A–3C**). No differences in genome:PFU ratios were observed between WT MAYV and MAYV E2-T179N during replication in Aag2 cells (**S5 Fig**).

Next, we assessed whether the E2-T179N mutation similarly impacts CHIKV, a closely related alphavirus that has caused major outbreaks globally in the last decades. Like MAYV, we observed that E2-T179N significantly enhanced CHIKV replication in Aag2 cells (**Fig 3D**; p = 0.0108 at 5 d.p.i.). However, in contrast to our MAYV results, we observed an increased replication in MRC-5 cells for the mutant compared to WT CHIKV (**Fig 3E**; p = 0.0164 at 2 d.p.i.). Together, these results indicate that E2 position 179 is a key residue impacting viral replication not only of MAYV but also of other related alphaviruses like CHIKV, albeit with different outcomes in human cells.

Considering the enhanced replication of MAYV E2-T179N in Aag2 cells, we hypothesized E2-T179N, which lies in the receptor binding domain of E2 domain B, may positively impact replication through increased binding of the mutant to Aag2 cells. To test our hypothesis, we performed binding assays of WT MAYV and MAYV E2-T179N with Aag2 cells. We observed significantly reduced binding of MAYV E2-T179N compared to WT (**S6 Fig**; p = 0.0005). This result suggested another component of the viral infection cycle may be impacted by E2-T179N such as egress since domain B of E2 plays a role in egress of CHIKV [42]. As such, we interrogated differences in release between WT MAYV and MAYV E2-T179N after the first round of replication at 12 h.p.i. by performing a one-step growth curve in Aag2 cells. Mutant titers were significantly higher than WT MAYV at 12 h.p.i (**Fig 3F**), and we observed an increase in the mutant titer by 9 h.p.i that was not observed in the WT virus even at 12 h.p.i. These data suggest that the mutant is released earlier from infected cells than WT MAYV.

## E2-T179N negatively impacts MAYV replication in human cells via Mxra8 and ApoER2

While the process by which MAYV enters insect cells remains largely unknown, there is evidence that entry in mammalian cells happens through the endosomal pathway [43] after

binding with Mxra8, originally identified as the receptor for CHIKV and several other alpha-viruses, including MAYV [44,45]. Thus, we investigated whether the negative effect of E2-T179N on viral replication in human cells (**Figs 2 and 3**) is due to decreased viral binding prior to entry and whether binding and replication is altered via the Mxra8 receptor and or other putative MAYV receptors.

To assess viral binding, we inoculated MRC-5 cells with either WT or MAYV E2-T179N at a MOI of 0.1 and incubated the cells at 4°C to allow for binding without internalization of the virus. We then measured the amount of bound virus via RT-qPCR, which showed significantly lower relative binding of MAYV E2-T179N compared to WT MAYV (~4.3 fold less on average, p<0.0001, **Fig 4A**).

To assess the potential impact of E2-T179N on MAYV E2 structure and binding to Mxra8, we used the cryo-EM structure of MAYV (PDB ID: 7KO8 [46]) aligned with the CHIKV structure in complex with Mxra8 (PDB ID: 6JO8 [47]). The E2-T179N change was not expected to have a major impact on the secondary structure of subdomain B or to impair protein-protein contacts between subdomains in the MAYV E2 (**S7A Fig**). However, the E2-T179 position is in proximity to the binding site for the Mxra8 receptor (**S7B Fig**). We thus reasoned that E2-T179N may impair the interaction with Mxra8, which may explain our results on binding to MRC-5 cells. To investigate the E2-T179N-Mxra8 interaction, we conducted several *in silico* analyses. First, we used two N-glycosylation prediction servers, NetNGlyc [48] and NGlycPred [49] to determine whether E2-T179N could create an additional N-glycosylation site via a N-X-S/T motif. Both methods failed to detect any additional N-glycosylation site in E2-T179N. Next, we used molecular modeling and molecular dynamics (MD) simulations with predicted free energy of binding calculations to evaluate if the E2-T179N mutation could affect the interaction between the E2 glycoprotein and the Mxra8 receptor. We obtained a $\Delta\Delta G$ of 0.15 ± 0.66 kcal/mol, indicating that the T179N mutation did not change predicted binding affinity to the Mxra8 D2 domain (p > 0.05, one-sample t-test) (**S7C Fig**). To consider small adjustments in the binding of Mxra8 to MAYV ectodomains, we modeled and performed molecular dynamics simulations in quintuplicate of the full MAYV spike, formed by three pairs of E1 and E2 ectodomains, in complex with 3 copies of the entire Mxra8 ectodomain. We calculated the binding free energy using MMGBSA and computed the difference between the relative binding free energy ($\Delta\Delta G$) of MAYV E2-T179N and WT MAYV spike to Mxra8. The $\Delta\Delta G$ predicted with MM/GBSA was –6.0 ± 26.3 kcal/mol (**S7D Fig**), which again indicates no significant (p > 0.05, one-sample t-test) free energy change associated with the E2-T179N mutation. Our structural analyses failed to detect any differences between binding of WT E2 and E2-T179N to Mxra8.

Given that structural analyses did not detect differences in E2-T179N binding to Mxra8, we sought to validate the results experimentally by performing binding assays in cells expressing Mxra8 and Mxra8 knock-out (KO) cells. WT MAYV and MAYV E2-T179N were inoculated onto WT 3T3 (mouse fibroblasts) and 3T3 Mxra8 KO cells at a MOI of 0.1, and the cells were incubated at 4°C. Unbound virus was washed away, and virus bound to the cells was quantified by RT-qPCR. We did not detect significantly less binding of MAYV E2-T179N to WT 3T3 cells as compared to WT MAYV, although there was a trend towards decreased binding which is similar to our binding results with MRC-5 cells (**Fig 4B**). However, both WT MAYV and MAYV E2-T179N bound less to 3T3 Mxra8 KO cells than to WT 3T3 cells. When we compared the reduction in binding to 3T3 Mxra8 KO cells from WT 3T3 cells between both viruses, we found no significant difference (**Fig 4C**). To assess whether E2-T179N alters replication through Mxra8, we performed growth curves of WT MAYV and MAYV E2-T179N in WT 3T3 and 3T3 Mxra8 KO cells at a MOI of 0.1, measuring infectious virus every 24 hours post-infection. Consistent with previous studies [44], replication of WT MAYV was decreased

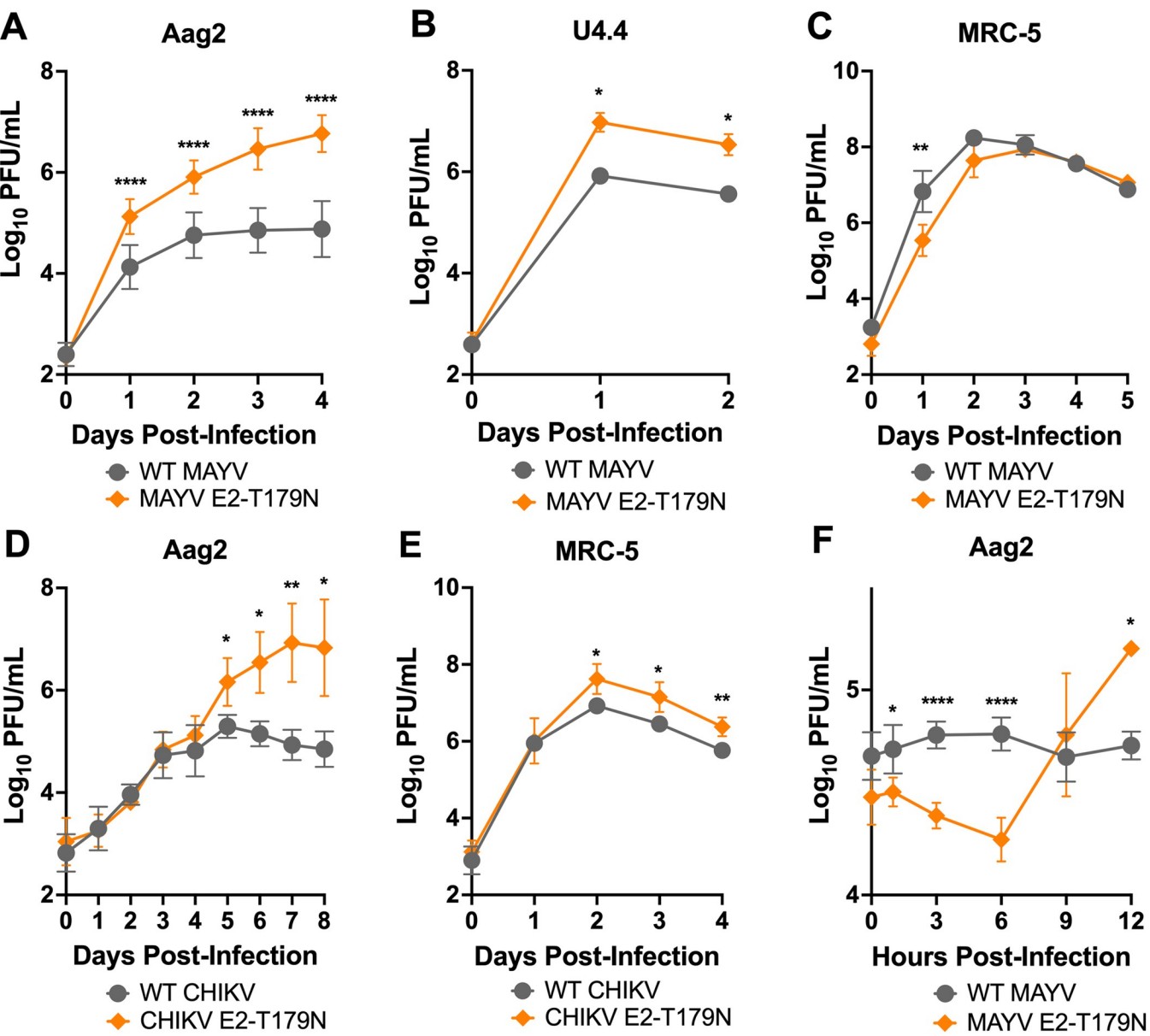

**Fig 3. E2-T179N increases viral replication of MAYV and CHIKV in insect cells and causes early release. A-C:** Replication of WT MAYV and MAYV E2-T179N in insect and human cells. Aag2 (**A**) and U4.4 (**B**) cells were infected with WT MAYV or MAYV E2-T179N at a MOI of 0.1, while MRC-5 were infected at a MOI of 0.01 (**C**). Replication was assessed over time by plaque assay titration. Data represents the mean of 2 independent experiments apart from the U4.4 growth curve which is representative of three independent experiments. **D-E:** Replication of WT CHIKV and CHIKV E2-T179N in insect and human cells. Aag2 cells were infected with WT CHIKV or CHIKV E2-T179N at a MOI of 0.1 (**D**), while MRC-5 were infected at a MOI of 0.01 (**E**). Replication was assessed over time by plaque assay. Data represents the mean of 2 independent experiments. **F:** One-step growth curve of WT MAYV and MAYV E2-T179N in insect cells. Aag2 cells were infected with WT MAYV and MAYV E2-T179N at an MOI of 10. Replication was assessed over time by plaque assay. Data represent the mean of 2 independent experiments. Statistical analysis: Two-Way ANOVA using Šídák's multiple comparisons; * = p<0.05; ** = p<0.01; **** = p<0.0001. Error bars represent the standard deviation.

in 3T3 Mxra8 KO cells compared to WT 3T3 cells at 1 and 2 d.p.i., indicating Mxra8 is important for replication (**Fig 4D**; p = 0.0095, p = 0.0007). Similarly, replication of MAYV E2-T179N was reduced at 1 and 2 d.p.i. in 3T3 Mxra8 KO compared to WT 3T3 cells (**Fig 4E**; p = 0.0126, p = 0.0485). To compare MAYV E2-T179N to WT MAYV, we calculated the mean change in $\log_{10}$-transformed virus titer between WT MAYV and MAYV E2-T179N in 3T3

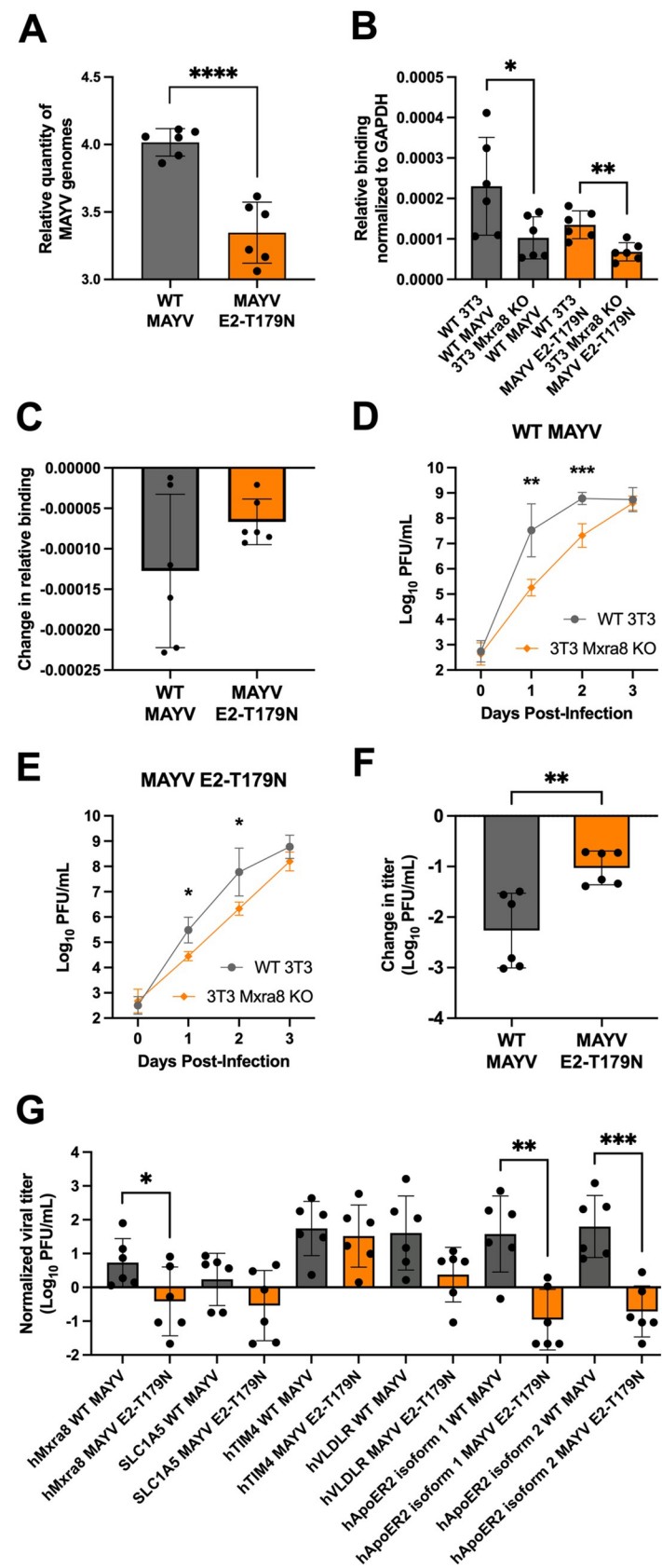

**Fig 4. Interaction with human receptors Mxra8 and ApoER2 may contribute to the attenuation of MAYV E2-T179N. A:** Binding of WT MAYV and MAYV E2-T179N to MRC-5 cells. Pre-chilled MRC-5 cells were infected at a MOI of 0.1, and virus was adsorbed to the cells at 4°C. Unbound virus was washed away, and bound virus was quantified by extracting RNA from the cells and performing RT-qPCR. Relative genome copies of bound virus were determined by normalizing the Ct value of bound virus to the Ct value of the housekeeping gene and virus in the inoculum. Statistical Analysis: unpaired t-test; **** = p<0.0001. Data comprise two independent binding assays, each with n = 3. The Y-axis is log-transformed for clearer visualization. **B-C:** Binding of WT MAYV and MAYV E2-T179N to WT 3T3 and 3T3 Mxra8 KO cells. Binding assays were performed as above for MRC-5 cells. The change in binding between WT 3T3 and 3T3 Mxra8 KO cells was calculated for each virus (**C**). Statistical Analysis: unpaired t-test; * = p<0.05; ** = p<0.01. **D-F:** Growth curves of WT MAYV and MAYV E2-T179N in WT 3T3 and 3T3 Mxra8 KO cell lines. WT 3T3 and 3T3 Mxra8 KO cells were infected at a MOI of 0.1 with either WT MAYV (**D**) or MAYV E2-T179N (**E**). Viral titers were quantified each day post-infection by plaque assay. Change in titer for each virus between WT 3T3 and 3T3 Mxra8 KO lines were compared (**F**). Data represent two independent biological replicates performed in triplicate. Statistical Analysis: Two-Way ANOVA with Šídák's multiple comparisons test and unpaired t-test; * = p<0.05; ** = p<0.01; *** = p<0.001. **G:** WT MAYV and MAYV E2-T179N infection in HEK-293A cells transiently expressing exogenous, putative MAYV receptors. Cells were transfected with constructs encoding hMxra8, hACE2, SLC1A5, hTIM4, hVLDLR, and hApoER2 isoforms 1 and 2. Twenty-four hours post-transfection, cells were infected with WT MAYV or MAYV E2-T179N at a MOI of 0.1. Infectious virus in the supernatant 24 h.p.i. was quantified by plaque assay and normalized to infection in cells expressing human ACE2. Data comprise two independent replicates, each performed in triplicate. Statistical Analysis: unpaired t-test; * = p<0.05; ** = p<0.01; *** = p<0.001. All error bars represent the standard deviation.

Mxra8 KO cells and WT 3T3 cells at 1 d.p.i. WT MAYV had a significantly greater reduction in titer in the absence of Mxra8 compared to the mutant (**Fig 4F**; p = 0.0039). This result suggests the reduced replication of MAYV E2-T179N in human cells is in part explained by its reduced dependence on Mxra8.

We explored other potential receptors that may explain additional mechanisms which attenuate MAYV E2-T179N in mammalian cells by infecting HEK-293A cells transfected with constructs encoding human SLC1A5, hTIM4, VLDLR, and ApoER2 isoforms 1 and 2. SLC1A5 is a component of the CD147 protein complex and was shown to be important for MAYV replication in HEK-293T cells, and the complex is implicated in the entry of CHIKV and other alphaviruses [50]. TIM4 is a human T-cell immunoglobulin and mucin-domain-containing protein which binds phosphatidylserine and was shown to enhance infection of Eastern equine encephalitis pseudovirus particles [51]. VLDLR and ApoER2 are receptors for many alphaviruses, including Semliki Forest virus, a close relative to MAYV [52]. As such, we hypothesized these putative MAYV receptors could contribute to differences in replication of WT MAYV and MAYV E2-T179N in mammalian cells. We transfected 293A cells with the constructs, including ACE2 and Mxra8 as negative and positive controls, respectively, and infected cells 24 hours later with either WT MAYV or MAYV E2-T179N at a MOI of 0.1. We measured replication by quantifying virus in the supernatant 1 d.p.i. After normalizing the data to replication in HEK-293A cells transfected with ACE2, we found that replication of MAYV E2-T179N was significantly decreased in cells expressing Mxra8 compared to WT MAYV (**Fig 4G**; p = 0.0476). Replication was also significantly increased for WT MAYV and MAYV E2-T179N in cells transfected with the construct encoding hTIM4 (p = 0.0031; p = 0.01), although there was no significant difference between replication of the two viruses. Compared to MAYV E2-T179N, ApoER2 isoforms 1 and 2, but not VLDLR, increased WT MAYV replication when exogenously expressed (p = 0.0016; p = 0.0004). These data indicate E2-T179N may negatively impact usage of the putative MAYV receptors ApoER2 isoforms 1 and 2, in addition to Mxra8, in mammalian cells.

## E2-T179N increases transmission potential for the urban vector *Ae. aegypti* after bypassing initial infection barriers

Given that the E2-T179N mutation increased fitness of MAYV in Aag2 cells, we hypothesized that it may impact MAYV transmission potential in *Ae. aegypti* mosquitoes. To test this

hypothesis, we performed five independent transmission studies across two study sites and two populations of *Ae. aegypti*: Kamphaeng Phet and Guerrero. We exposed groups of *Ae. aegypti* to an artificial bloodmeal containing either WT or MAYV E2-T179N. To assess infection rates, we collected either midguts, as this is the first site of infection, or bodies lacking legs and wings. In our studies with the Kamphaeng Phet population, both WT and MAYV E2-T179N infected mosquitoes to high rates (**Fig 5A**), although WT MAYV infection rates were higher than for MAYV E2-T179N (p = 0.035). In the Guerrero population, WT MAYV infection rates were also significantly higher than MAYV E2-T179N (**Fig 5B**; p = 0.0035).

We collected legs and wings to assess the ability of MAYV to disseminate to peripheral organs and did not observe significant differences between WT and MAYV E2-T179N in the Kamphaeng Phet population (**Fig 5A**; p = 0.093) but did observe significantly lower dissemination rates for MAYV E2-T179N in the Guerrero population (**Fig 5B**; p = 0.0034). In contrast, we observed a significant increase in the proportion of mosquitoes able to transmit MAYV E2-T179N in the Kamphaeng Phet population (**Fig 5A**; p<0.0001), but transmission rates were similar in the Guerrero population (**Fig 5B**).

Since several transmission barriers exist within the mosquito [53], we calculated dissemination and transmission rates based on the mosquitoes that became infected (**Fig 5C–5F**) or those with a disseminated infection (**Fig 5G and 5H**) to understand how each barrier contributes to transmission of MAYV E2-T179N. The rate of infected mosquitoes that developed a disseminated infection was similar for both viruses (**Fig 5C and 5D**), indicating that the viruses had similar ability to disseminate from the midgut after successful infection of the midgut. The transmitting of infected rate was higher for MAYV E2-T179N compared to WT MAYV in the Kamphaeng Phet population (**Fig 5E**; p<0.0001), and the Guerrero population trended toward a similar result (**Fig 5F**; p = 0.0731). Finally, we calculated the transmission rates for mosquitoes with a disseminated infection and found significantly higher rates for MAYV E2-T179N than WT MAYV in both the Kamphaeng Phet (**Fig 5G**; p<0.0001) and Guerrero populations (**Fig 5H**; p = 0.0407). Our results indicate MAYV E2-T179N is less able to overcome the initial midgut infection barrier; however, once MAYV E2-T179N has disseminated, it is more likely to enter the saliva than WT MAYV.

## MAYV E2-T179N generates reduced viremia and tissue pathology in mice

Since MAYV E2-T179N exhibited increased replication in insect cells but reduced replication in mammalian cells compared to WT MAYV, we sought to explore whether adaptation to *Ae. aegypti* comes at the cost of reduced viral fitness in a mammalian host. Thus, we infected CD-1 mice with WT MAYV and MAYV E2-T179N. Mice infected with MAYV E2-T179N had significantly lower viremia than mice infected with WT MAYV at all timepoints tested (**Fig 6A**) but mice consistently gained weight over the course of the study in both groups (**Fig 6B**). Mice infected with MAYV E2-T179N had significantly lower footpad swelling relative to WT MAYV at five days post infection (**Fig 6C**; p = 0.0006) and delayed peak swelling, although swelling between the groups was similar. To determine whether MAYV E2-T179N infection caused less tissue damage compared to WT MAYV, we collected the inoculated footpad of each mouse at seven days post-infection and processed them for histological analysis. Based on the severity of lymphoplasmacytic myositis, scores ranging from zero to three were assigned to samples, with zero denoting normal tissue and three the greatest extent of myofiber loss across all samples. Mice infected with MAYV E2-T179N yielded significantly lower footpad scores than those infected with WT MAYV (**Fig 6D**; p = 0.0150), indicating lower levels of tissue damage at this timepoint. Images taken from the stained cross-sections of footpads show the degree of muscle fiber degeneration and immune cell infiltration, with this symptom being

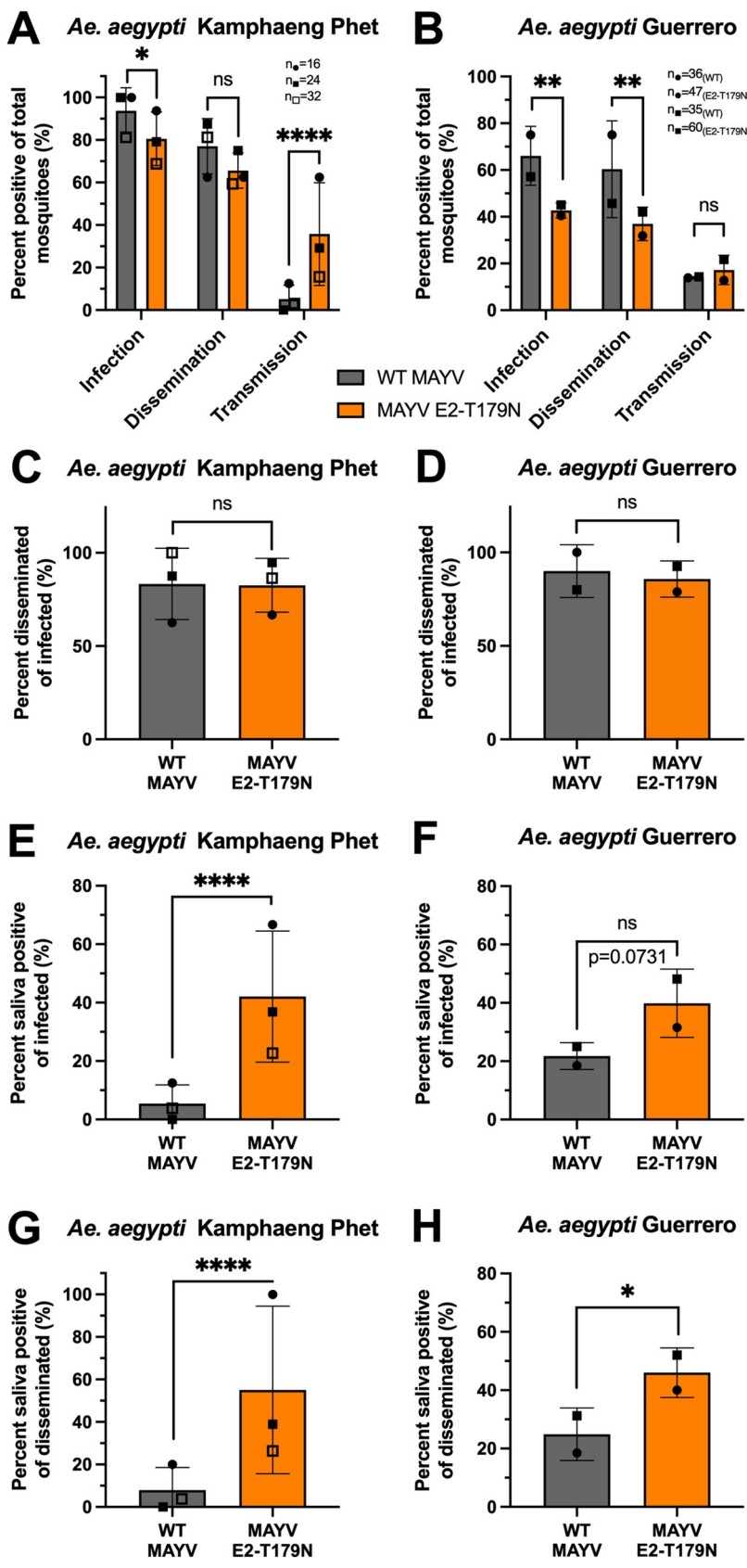

**Fig 5. Vector competence of *Ae. aegypti* infected by WT and MAYV E2-T179N. A-B:** Adult *Ae. aegypti* Kamphaeng Phet colony mosquitoes were exposed to a bloodmeal containing $10^6$ PFU/mL (**A**) and *Ae. aegypti* Guerrero colony mosquitoes exposed to a bloodmeal containing $10^7$ PFU/mL (**B**) of either WT MAYV or MAYV E2-T179N. Fully engorged mosquitoes were kept at 28°C with 80–90% relative humidity, 12hr light:12hr dark, and permanent access to a 10% sucrose solution. Mosquitoes were cold-anesthetized and dissected at 7 d.p.i. (**A**) or 10 d.p.i. (**B**). Viral titers in the midgut (**A**) or bodies (**B**) and legs and wings were quantified by plaque assay titration. During dissection, saliva was harvested from live mosquitoes and virus was either amplified on C6/36 or BHK-21 cells (**A**). After 3 days, presence of virus was detected using RT-qPCR or direct visualization of CPEs. For the experiments with *Ae. aegypti* Guerrero mosquitoes, virus in the saliva was quantified by plaque assay, and plaques as little as one in neat saliva were validated by RT-qPCR (**B**). The infection, dissemination, and transmission rates are represented as positive mosquitoes of total bloodfed mosquitoes and span 3 independent experiments (with n = 16, n = 24, and n = 32 per group for the first, second, and third replicates, respectively; n = 72 total/group) are represented as positive mosquitoes of total bloodfed mosquitoes (**A**). Infection, dissemination, and transmission rates are also represented as positive mosquitoes of total bloodfed mosquitoes and span 2 independent experiments (with n = 36/n = 47 and n = 35/n = 60 for WT/mutant-infected groups for replicates one and two, respectively; total n = 71 for WT and total n = 107 for mutant) (**B**). **C-D:** Percentage of mosquitoes positive for virus in legs and wings of infected mosquitoes as measured by RT-qPCR on amplified saliva samples (**C**) or plaque assay and RT-qPCR (**D**). **E-F:** Percentage of mosquitoes positive for virus in saliva of infected mosquitoes as measured by RT-qPCR on amplified saliva samples (**E**) or plaque assay and RT-qPCR (**F**). **G-H:** Percentage of mosquitoes positive for virus in saliva of mosquitoes with disseminated virus in legs and wings as measured by RT-qPCR on amplified saliva samples (**G**) or plaque assay and RT-qPCR (**H**). Statistical analysis: ns = not significant, * = p<0.05; ** = p<0.01; **** = p<0.0001 (Two-tailed Fisher's exact test). Sample sizes for independent experiments are described panels A and B and relate to unique symbols for data points on the graphs. All error bars represent the standard deviation.

more severe in the group infected with WT MAYV (**Fig 6E, panel 1**) than MAYV E2-T179N (**Fig 6E, panel 2**). The primary infiltrating immune cells were lymphocytes, and to a lesser extent, plasma cells and macrophages. A representative image from mock-infected mice is presented in **Fig 6E, panel 3**. These results demonstrate that MAYV E2-T179N replicates to lower levels and causes less severe tissue pathology compared to WT MAYV in the mammalian host.

## Discussion

MAYV is an emerging viral threat with epidemic potential. The main goal of this study was to determine whether evolution of MAYV may lead to viral emergence through adaptation to the urban vector *Ae. aegypti*. We identified E2-T179N as a mutation that confers increased replication in *Ae. aegypti* cells (**Figs 2 and 3**) and enhances transmission rates after initially overcoming the midgut infection barrier in two distinct populations of *Ae. aegypti* (**Fig 5**). Thus, we demonstrate MAYV has the potential to adapt to *Ae. aegypti* and enter the urban transmission cycle in some capacity.

Transmission studies of WT MAYV and MAYV E2-T179N in *Ae. aegypti* Kamphaeng Phet and Guerrero populations demonstrated a reduced infection rate for MAYV E2-T179N, however this was paired with an increased transmission rate of mosquitoes with a disseminated infection compared to WT MAYV. However, we observed similar dissemination rates and a trend toward higher transmission potential of the mosquitoes that became infected. These data suggest E2-T179N has an initial fitness reduction at the midgut which is later overcome as the virus either infects and or escapes the salivary gland. The mechanism by which this occurs should be explored further by investigating differences in receptor populations across mosquito tissue types, and the translatability of these data to other alphaviruses such as CHIKV should also be explored.

The same E2-T179N mutation caused a fitness reduction in human cells (**Figs 2 and 3**) and mice (**Fig 6**), suggesting that this mutation alone would not favor a complete, urban MAYV transmission cycle. Other roadblocks remain in the way of MAYV emergence: notably, it is still unclear whether the viremia induced by MAYV in humans is sufficient to sustain human-to-human amplification. With reduced viremia in mice (**Fig 6A**), it is even less likely MAYV

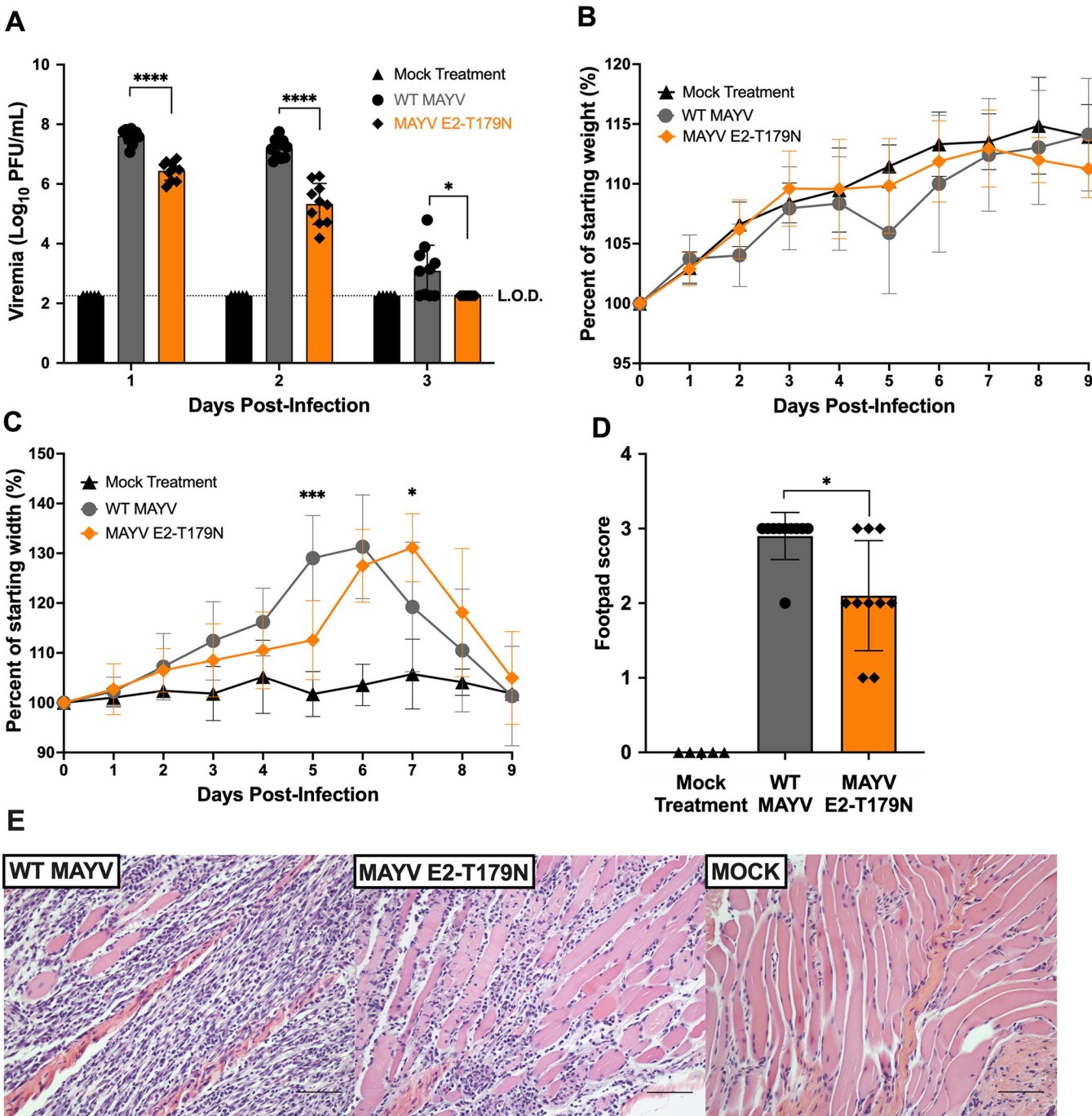

**Fig 6. E2-T179N attenuates MAYV replication and tissue pathology in mice.** Four-week-old CD-1 mice were infected with $10^5$ PFU in 50 μL via injection to the left, hind footpad. **A:** Viral load in serum was assessed daily by plaque assay. Statistical comparisons were made using multiple unpaired t tests with correction for multiple comparisons using the false discovery rate method of Benjamini, Krieger, and Yekutieli. **B**: Mice were weighed daily following infection to assess weight change. **C:** Footpad swelling was measured daily using a digital caliper. **B-C:** Statistical comparisons were made using a two-way ANOVA with Dunnett's correction for multiple comparisons. Data represent the percent of the weight or footpad width before infection. **D:** Footpad swelling score as determined by histopathology whereby a score of 0 indicates a normal state, 1 mild (<25%) myofiber loss, 2 moderate (25–50%) myofiber loss, and 3 severe (>50%) myofiber loss. Statistical comparisons were made by Mann-Whitney test. **E:** Hematoxylin and eosin stain of muscular tissue in left, hind footpad of mice seven days post infection with WT MAYV (left), MAYV E2-T179N (center) or after a mock treatment (right). Footpad scores of 3 and 1 are shown for images of WT MAYV and MAYV E2-T179N, respectively. The scale bar in the images is 200 microns. Studies were performed in two independent experiments, using 10 mice per infected group and five mock-treated mice. Error bars represent standard deviation from the mean.

E2-T179N generates enough viremia for human transmission. Serum viral loads of $10^4$–$10^5$ PFU/mL MAYV have been observed in a study involving 21 humans with confirmed MAYV infection during the three days following symptom onset [4]. In comparison, infectious titers for CHIKV range from $10^{3.9}$ to $10^{6.8}$ PFU/mL [54]. Thus, titers in humans appear to be slightly lower for MAYV than the successful human pathogen, CHIKV, which may limit emergence potential without further adaptation. Furthermore, the geographical overlap between MAYV and CHIKV may prevent MAYV emergence, as studies in mice have suggested that a prior CHIKV infection confers cross-protection against MAYV infection [55,56].

The two different experimental evolution approaches we used (serial passaging and DMS) led to the identification of E2-T179N as a key mutation enhancing MAYV replication in *Ae. aegypti* and *Ae. albopictus* cells. Notably, we observed that only three serial passages of DMS viruses in Aag2 cells allowed identification of this adaptive mutation (**Fig 1F**), while we observed this mutation in only 50% of the replicates in our serial passaging experiment after ten passages (**Fig 1C**). This confirms that DMS passaging speeds up the process of selection, although DMS can only be targeted to a portion of the viral genome. Thus, we conclude that combining these two approaches represents an efficient strategy to identify adaptive mutations with high confidence. Perhaps unsurprisingly, most mutations detected with both approaches occurred within the receptor-binding protein (E2), highlighting the importance of this protein for host range. Furthermore, the residues with the strongest impact on fitness (positions 179 and 183 in E2) both reside within domain B, which we previously showed was critical for cell tropism [57]. A surprising finding in our DMS studies was that variants enriched during passage in LLC-MK2 either had neutral or deleterious impacts on fitness in mammalian cells. However, these data are consistent with results observed in ZIKV [38] where mammalian fitness gains were only observed with a double mutant. A possible explanation for this is the importance of cooperative interactions with other residues (either on the same genome or on other genomes) occurring within the viral population. Moreover, a mutation that was slightly enriched during mouse passage (E2-I354D) decreased fitness appreciably in both mammalian and insect cells; thus, this mutation represents a broadly attenuating mutation and suggests that DMS studies may be useful for developing rationally designed attenuated vaccine viruses. For example, in future studies, selecting the residues most differentially selected against may lead to a highly attenuated virus and could help increase understanding of fundamental virus biology using a safer loss-of-function approach.

Given the reduced fitness, replication, and binding of MAYV E2-T179N to human fibroblasts MRC-5, we sought to explore known and putative human MAYV receptors which may be mediating these differences. Binding assays to WT 3T3 and 3T3 Mxra8 KO cells showed no difference in the reduction in binding posed by the loss of Mxra8 between the viruses, however, there was a larger reduction in replication of WT MAYV than MAYV E2-T179N in 3T3 Mxra8 KO cells. We conclude the human MAYV receptor Mxra8 contributes to the replication attenuation of MAYV E2-T179N in mammalian cells.

To date, there is no evidence TIM-family receptors, VLDLR, and ApoER2 are used specifically by MAYV. We tested a panel of putative MAYV receptors by infecting HEK-293A cells transiently expressing exogenous human TIM4, VLDLR, and ApoER2 isoforms 1 and 2 with WT MAYV and found increased replication in the presence of these host factors. These results warrant further studies to elucidate the precise role they play in infection. MAYV E2-T179N displayed significantly decreased replication than WT MAYV in the presence of exogenous Mxra8 and ApoER2 isoforms 1 and 2 (**Fig 4G**). While there may be other factors responsible for the attenuation of MAYV E2-T179N such as binding to heparan sulfate [58], these data suggest that MAYV E2-T179N's attenuation may in part be mediated through interactions with these host factors.

While we showed E2-T179N negatively impacts binding and viral replication in MRC-5 cells, and replication through the human receptors Mxra8 and ApoER2, the mechanisms underlying the impact of E2-T179N on binding and replication in Aag2 cells are still unclear. The significantly reduced binding of MAYV E2-T179N to Aag2 cells was unexpected given its increased replication in these cells. We hypothesize E2-T179N compensates reduced binding via increased fitness later in the viral life cycle. Our data showed MAYV E2-T179N is released earlier than WT MAYV in Aag2 cells (**Fig 3F**), suggesting there may be differences in egress or some other stage during replication. This may be through interactions with other viral proteins; for example, it is known that the cytoplasmic domain of E2 of alphaviruses such as Sindbis virus interact with the nucleocapsid core to promote budding [59]. Structural analyses of E2-T179N in complex with MAYV nucleocapsid could elucidate changes to E2 that may positively impact encapsidation and budding efficiency. Other possible explanations for the phenotypes observed are disrupted E1:E2 interactions, since this residue resides in domain B, which interacts with E1 and could disrupt endosomal fusion [60]. Future studies will help shed light on the mechanisms responsible for the phenotypes observed.

Alphaviruses, like other arboviruses, navigate evolutionary trade-offs because of their dual-host nature. Our studies exemplified this concept: MAYV cannot optimize its replication in both insect vectors and mammalian hosts simply through the E2-T179N mutation, possibly due to the need to use different receptors in each host or even that a mutation in one environment may increase replication through efficient egress while simultaneously negatively impacting receptor binding in another host. Mutations well-suited for increased fitness in one part of the viral infection cycle may negatively influence fitness in another part of the cycle or within or between hosts. However, it is possible that additional mutations in E1 or E2 may compensate for this defect, and/or that evolution at the T179 towards other residues may confer increased replication in both hosts. Interestingly, we observed that two distinct mutations (T179N and T179I) were enriched in mosquito cells. Therefore, testing other T179 mutation combinations may provide additional insights into MAYV viral evolution potential. A limitation of this work is our use of a single strain and genotype of MAYV, and future studies should assess the impact of E2 T179N on other MAYV genotypes. Importantly, previous reports have shown a viral strain-dependent impact of the CHIKV A226V mutation that enhances *Ae. albopictus* transmission for the ECSA genotype, but not the Asian genotype [61]. To that end, we found that E2-T179N CHIKV mutant similarly displayed increased fitness in *Ae. aegypti* cells (**Fig 3D**), but, in contrast to MAYV, also showed increased fitness in human fibroblasts (**Fig 3E**). Thus, this residue is important for fitness for both viruses but the differential impact on fitness suggests different mechanisms or epistatic interactions. This result is consistent with the effect we observed with the E1-A226V MAYV mutant, which is known to enhance CHIKV's fitness in *Ae. albopictus* [62] but greatly reduced MAYV's fitness in U4.4 cells (**Fig 2C**). Finally, given that E2-T179N in MAYV did not alter fitness in monkey cells (**Fig 2D**), we cannot conclude that this is a mammalian- or insect-specific phenotype; rather, it appears to be species-specific, likely dependent on the presence of various entry factors and receptors.

In addition, the E2-T179N mutation led to reduced viral replication and tissue damage *in vivo* in a mouse model, which was consistent with the attenuation of MAYV E2-T179N in human and mouse fibroblasts (**Figs 3 and 4**). Future work will establish whether reduced tissue pathology of MAYV-induced disease is solely due to lower viral replication, or to other immune parameters such as inflammation, cytokine response, IFN signaling, etc. It will also be important to study why no differences in footpad swelling were observed despite differences in tissue damage, which may shed light on the inflammatory processes induced by alphaviruses. Furthermore, future studies will assess whether viremia levels in mice are still sufficient to cause productive infection in mosquitoes. Our work suggests that mutations in the E2 of

MAYV may favor its dissemination through the usage of alternate mosquito vectors and should therefore be monitored closely with appropriate surveillance programs.

## Materials and methods

### Ethics statement

**Human blood and ethics statement.** Human blood used to feed mosquitoes was obtained from healthy volunteer donors. Healthy donor recruitment was organized by the local investigator assessment using medical history, laboratory results, and clinical examinations. All adult subjects provided written informed consent. Biological samples were supplied through participation of healthy volunteers at the ICAReB biobanking platform (BB-0033-00062/ICAReB platform/Institut Pasteur, Paris/BBMRI AO203/[BIORESOURCE]) of the Institut Pasteur to the CoSImmGen and Diagmicoll protocols, which have been approved by the French Ethical Committee (Comité de Protection des Personnes; CPP) Ile-de-France I. The Diagmicoll protocol was declared to the French Research Ministry under the reference: DC 2008–68 COL 1.

**Biosafety protocols.** The research protocols at Virginia Tech were approved by the Institutional Animal Care and Use Committee (IACUC #18–084) and the Institutional Biosafety Committee of Virginia Tech (IBC #18–026). All studies performed at the Institut Pasteur were approved by the Committee of Health, Safety, and Working Conditions under protocol number 75–1448. Biosafety practices at Virginia Tech were conducted according to the recommendations of the National Institutes of Health and the Centers for Disease Control and Prevention which included performing mosquito infections at ACL3 and mouse infections at ABSL2. The mouse studies were carried out in strict accordance with the recommendations in the Guide for the Care and Use of Laboratory Animals of the National Institutes of Health. Handling of MAYV and CHIKV was performed at BSL2 and BSL3, respectively. At Institut Pasteur, all work with MAYV was performed in dedicated BSL3 facilities.

### Cells, viruses and plasmids

Vero, BHK-21, LLC-MK2, and MRC-5 cells were grown in Dulbecco's modified Eagle's medium (DMEM, Gibco), containing 10% fetal calf serum (FBS; Gibco), 1% penicillin/streptomycin (P/S; Thermo Fisher) in a humidified atmosphere at 37˚C with 5% $CO_2$. U4.4 and Aag2 cells were maintained in Leibovitz's L-15 medium (Gibco) with 10% FBS, 1% P/S, 1% non-essential amino acids (Sigma) and 1% tryptose phosphate (Sigma) in a dry atmosphere without $CO_2$ at 28˚C. 3T3 WT and Mxra8 KO cells were a kind gift of Dr. Michael Diamond and were maintained in a humidified atmosphere at 37˚C with 5% $CO_2$ in DMEM (Genesee Scientific) supplemented with 10% fetal bovine serum (R&D Systems), 1% non-essential amino acids, 50 µg/mL gentamicin sulfate, and 25 mM HEPES. HEK-293A cells were grown in DMEM (Genesee Scientific) containing 5% FBS (R&D Systems), 1% non-essential amino acids, 50 µg/mL gentamicin sulfate, and 25 mM HEPES in a humidified atmosphere at 37˚C with 5% $CO_2$. For some studies, Aag2 were maintained in Schneider's insect medium (Genesee Scientific) with 7% FBS, 1% non-essential amino acids, 2.5 µg/µL Amphotericin B, 50 µg/mL gentamicin sulfate, and 5% tryptose phosphate broth in a humified atmosphere without $CO_2$ at 28˚C. MAYV strain TRVL 4675 was derived from an infectious clone we previously described that contains an SP6 promoter to generate infectious RNA [63]. To generate the MAYV infectious clones under the CMV promoter, we amplified the entire MAYV genome from the previously described SP6-containing plasmid along with the pcDNA3.1-based CMV vector [41] containing overlapping ends. All clone sequences were confirmed using next generation sequencing (NGS). The generation of the infectious molecular clone for CHIKV strain SL-CK1 was described previously [41]. The expression construct encoding hTIM4 was provided by H.

Choe [51], constructs for hVLDLR, hApoER2 isoform 1, hApoER2 isoform 2, and hMxra8 were provided by J. Abraham [52], and constructs for expression of hACE2 receptor were provided by G. Larson from the United States Centers for Disease Control and Prevention. The pLX304 vector containing SLC1A5 was obtained from the Arizona State University DNASU Plasmid Repository.

## Viral stocks

To generate MAYV viral stocks for the serial passaging experiments, a T25 flask of BHK-21 cells at 75% confluency was transfected with 10 μg of MAYV CMV-promoter driven plasmid using TransIT-LT1 Transfection Reagent (Mirus) according to the manufacturer's instructions. Viral supernatants were collected 48h later and used to perform one blind passage on C6/36 cells. The high titer viral stock used for viral growth curves and mosquito experiments was generated the same way, except that the virus was passaged twice on Vero cells (MOI 0.01). The generation of MAYV and CHIKV mutant stocks was performed by transfecting 500 ng of DNA into a 24-well plate well using JetOptimus (Polyplus, France). The mutants were then passaged once in BHK-21 cells at a MOI of 0.01. Viral stocks were titrated by plaque assay.

## Plaque assay

Vero cells were seeded in 24-well plates ($10^5$ cells per well) and infected with serial dilutions of infectious supernatant diluted in RPMI-1640 with 2% FBS and 10 mM HEPES (viral diluent). After 1h at 37˚C, a semi-solid overlay consisting of 1.5% methylcellulose, 2X EMEM, 4X L-glutamine, 0.735% sodium bicarbonate, 0.2 mg/mL gentamicin sulfate, 4% heat-inactivated FBS, and 20 mM HEPES was added to the cells. Cells were fixed with 4% formalin solution, pH 6.2 at 3 or 4 d.p.i. and stained with 0.1% crystal violet.

## Viral growth curves

One or two days prior to infection, cells were seeded in a 24-well plate and infected at 60–80% confluency. Cells were infected in triplicate at a MOI of 0.01 for all cells except Aag2, WT 3T3, and 3T3 Mxra8 KO cells, which were infected at MOI 0.1. Virus was diluted in RPMI-1640 (Genesee Scientific) media containing 2% FBS and 10 mM HEPES. One hour post-infection, viral inoculum was removed, cells were washed once with PBS, and the appropriate culture media was added to the cells. Cells were incubated at either 37˚C or 28˚C, according to cell type and conditions listed above. Supernatant was collected every 24 hours for the indicated time points and frozen until viral titration. One-step growth curves were performed by infecting Aag2 cells at a MOI of 10. Cells were chilled at 4˚C for 30 minutes and adsorption was performed at 4˚C for 30 minutes to synchronize infection. Unbound virus was washed away with cold PBS six times, and internalization and infection were initiated by adding media to cells and incubating at 28˚C. Supernatants were harvested every 3 hours post-infection, and all infectious virus was quantified by plaque assay.

## Serial passaging

We infected cells at a MOI of 0.01 as indicated above for viral growth curves, except infections were carried out with 6 replicates instead of 3. Viral supernatants were collected (2 days p.i. for BHK-21 cells; 3 days p.i. for U4.4 and Aag2 cells) aliquoted, and frozen. After each passage, all supernatants were titrated by plaque assay. Supernatants collected at passages 1, 5 and 10 were sequenced using NGS as described below.

## NGS for serial passaged samples

RNA of 100 μL of each sample supernatant was extracted using TRIzol reagent (Invitrogen) following the manufacturer's protocol. RNA was resuspended in 30 μL of nuclease-free water. After quantification using Quant-IT RNA assay kit (Thermo Fisher Scientific), viral RNA was enriched using polyA selection NEBNext Poly(A) mRNA Magnetic Isolation Module (NEB). Libraries were prepared with NEBNext Ultra II RNA Library Prep Kit for Illumina. The quality of the libraries was verified using a High Sensitivity DNA Chip (Agilent) and quantified using the Quant-IT DNA assay kit (Thermo Fisher Scientific). Sequencing of the libraries was performed on a NextSeq 500 sequencer (Illumina) with a NextSeq Mid Output kit v2 (151 cycles).

## Generation of MAYV envelope DMS populations

DMS populations were created in the genes encoding E3, E2, 6K, and E1 using the SP6-driven MAYV infectious clone as previously described [36–38] with modifications. Notably, we used a bacteria-free cloning approach that we have previously described [64,65]. DMS mutagenesis primers are listed in **S4 File**. The forward mutagenesis primers pool was used with a corresponding MAYV reverse end primer (5' CCCGCATTACACGGTACTTATGAT 3'). The reverse mutagenesis primers pool was used with a corresponding MAYV forward end primer (5' CGGAAGGCACAGAGGAGTGG 3'). We performed one round of mutagenesis with ten PCR cycles. All PCRs were performed with SuperFi II PCR master mix (Invitrogen). The fragments were then joined by PCR. The vector containing the remaining MAYV genome was amplified to create overlapping ends with the mutagenized fragments that are compatible with Gibson assembly. The primer sequences used to amplify the vector are forward (5' CCACT CCTCTGTGCCTTCCG 3') and reverse (5' ATCATAAGTACCGTGTAATGCGGG 3'). Amplicons were run on a 0.6% agarose gel containing GelGreen nucleic acid stain, excised, and then purified using the NucleoSpin Gel and PCR clean-up kit (Macherey-Nagel). Amplicons were then assembled (1:1 insert:vector molar ratio) into circular molecules using the NEBuilder HiFi DNA Assembly Master Mix incubated at 50˚C for two hours. To confirm that no parental plasmid vector was carried through the process, for each mutant, we included a control containing the DNA fragments but no assembly mix; this was then treated identically to the other samples for the remainder of the process. The assembly was then digested with exonuclease I, lambda exonuclease, and DpnI (all from NEB) to remove single-stranded DNA, double-stranded DNA, and bacterial-derived plasmid DNA, respectively. This product was then amplified by rolling circle amplification (RCA) using the Repli-g mini kit (Qiagen). The RCA product was linearized with SgrAI (NEB) and purified. Capped RNA was generated using the mMESSAGE mMACHINE SP6 kit (Invitrogen) and then transfected into a T150 flask of BHK-21 cells using JetMessenger (Polyplus). Viral supernatants were harvested 48h later, clarified by centrifugation, and then precipitated using polyethylene glycol (PEG) 8000 [66]. We constructed three replicate libraries by performing all the steps independently for each replicate starting with individual PCR reactions.

## DMS passaging

Passaging *in vitro* was performed in Aag2 and LLC-MK2 at a MOI of 0.01 for a total of three passages. Following one passage, the virus was titrated by plaque assay and the next passage was initiated at the same MOI. Only one passage was performed for *in vivo* studies. Three week-old female C57BL/6J mice (Jackson laboratory) were inoculated with $10^6$ PFU of WT MAYV or DMS populations in the left hind footpad. The mice were bled on days one and two post-infection to measure viremia and collect virus for sequencing. *Ae*. *aegypti* females originally collected in Guerrero (Mexico) were exposed to an infectious bloodmeal containing

$3 \times 10^7$ PFU/mL. Fully bloodfed mosquitoes were separated and allowed to incubate at 28˚C for ten days. Following incubation, salivary secretions were collected in viral diluent containing RPMI-1640 media supplemented with 25 mM HEPES, 1% BSA, 50 μg/mL gentamicin, and 2.5 μg/mL amphotericin B. We collected only saliva samples because this best represents the virus with potential to transmit to the mammalian host.

### DMS sequencing and analysis

Viral RNA for sequencing was extracted using the Direct-zol RNA extraction kit (Zymo Research). RNA was converted to cDNA using the Maxima H Minus cDNA Synthesis Master Mix (Invitrogen). PCR amplicons containing the DMS region were amplified using SuperFi II master mix with MAYV specific primers: forward (5' AGTGGGTAAGCCTGGCGACA 3') and reverse (5' TAAATCGGTCCGCATCATGCAC 3'). The fragments were purified using a 1x ratio of AMPure XP (Beckman). Libraries were prepared using Nextera XT and sequenced on an Illumina NextSeq 500. Data was analyzed using the dms_tools2 software, which is available at https://jbloomlab.github.io/dms_tools2/. Differential selection was calculated by comparing the starting virus stock to the post-passage samples.

### NGS data analysis

Raw sequencing data was deposited under the Sequence Read Archive (SRA, NCBI) under bioproject number PRJNA796940. The raw NGS reads were first trimmed using BBDuk, a tool from the BBMap tool kit, to remove the sequencing adapters and reads with a quality score of less than 30 [67]. The sequences were then aligned to the MAYV genome using the Burrow Wheeler Aligner (BWA) tool with no added parameters [68]. The resulting.bam files were then sorted using Sambamba [69]. Indels and variants were called using LoFreq [70] and then filtered to remove variants present at less than 5% frequency. We then used snpdat to annotate the variants [71]. A variant threshold was calculated by comparing the coverage depth to the allele frequency. Specifically, we took the reciprocal frequency of a given variant and multiplied by 10; if the sequencing coverage at this site was greater than that number, it was considered positive; otherwise, it was discarded [72].

### Site-directed mutagenesis and virus rescue

All mutants were created using our bacteria-free cloning approach described above with some modifications. The CMV-promoter-driven plasmids for MAYV and CHIKV were used to make mutants. Mutations were incorporated into PCR primers that created 20–30 bp overlaps for Gibson assembly; all primer sequences are available upon request. PCR fragments were amplified using Platinum SuperFi II PCR master mix (Invitrogen). The assembled product was amplified by RCA with the FemtoPhi DNA amplification kit (Evomic Science). Virus rescue was performed by transfecting RCA products directly into BHK-21 cells using JetOptimus (Polyplus), as we have previously described [73]. Viral titers were measured by plaque assay. To sequence the viral genome, we extracted RNA, performed DNase treatment to remove residual plasmid DNA, and generated cDNA using Maxima RT (ThermoFisher). PCR amplicons were generated using SuperFi II PCR master mix. Following gel or PCR purification, amplicons were submitted for Sanger sequencing at the Virginia Tech Genomics Sequencing Center.

### Genome:PFU ratio

A viral growth curve was performed in Aag2 cells, as above, with a MOI of 0.1 for both WT MAYV and MAYV E2-T179N. Supernatant was harvested each day post infection and titrated

by plaque assay. Viral genomes were quantified each day post infection via RT-qPCR on a Bio-Rad CFX96 Touch Real-Time PCR Detection System. Briefly, supernatant was diluted 1:5 in nuclease-free water before performing reverse transcription using the New England Biolabs Universal One-Step RT-qPCR kit followed by SYBR green-based qPCR using the following primers: MAYV 5028For. (5' CCTCTGTTAGTCCTGTGCAATAC 3'); MAYV 5108Rev. (5' AAGGTGCTTAGGGAGCTACT 3'). A standard curve was generated using full-length MAYV genomic RNA derived from a SP6-containing MAYV infectious clone [73] to calculate MAYV genomes per milliliter of supernatant, and the genome:PFU ratio was calculated by dividing the genome concentration by the PFUs per milliliter of supernatant achieved via pla-que assay.

## Competition assays

*In vitro* competition assays were performed against a genetically marked reference virus essentially as previously described [39–41]. Briefly, the competitor and test virus (in this case WT or mutant MAYV) were mixed at an equal PFU:PFU ratio and then used to infect cells at a MOI of 0.01. Virus-containing supernatant was harvested at two days p.i. for all cell lines used: Aag2, U4.4, MRC-5 and LLC-MK2. The proportion of each virus was measured using TaqMan RT-qPCR with probes specific to each virus containing ZEN / Iowa Black FQ quenchers. The primers used are as follows: MAYV 5028For. (5' CCTCTGTTAGTCCTGTGCAATAC 3'); MAYV 5108Rev. (5' AAGGTGCTTAGGGAGCTACT 3'); WT probe (5' FAM CACAGTGA AACTACTGTAAGCTTGAGCTCG 3'); Competitor probe (5' JOE CACAGTGAAACTACT GTttcCcTttcCTCG 3'). The genome copies of each virus in a given sample were calculated using a standard curve of viral RNA derived from the SP6 promoter-based clones for both WT and the marked reference virus. The relative fitness was determined with methods previously described [39,74] using the genome copies for each virus. Briefly, the formula $W = [R(t)/R(0)]^{1/t}$ represents the fitness (W) of the mutant genotype relative to the common competitor virus, where $R(0)$ and $R(t)$ represent the ratio of mutant to competitor virus in the inoculation mixture and at $t$ days post-inoculation, respectively.

## Binding assays in MRC-5 and Aag2 cells

Pre-chilled MRC-5 or Aag2 cells were washed with 4°C PBS containing 5% w/v BSA prior to inoculation with WT MAYV or MAYV E2-T179N at a MOI of 0.1. Adsorption of the virus proceeded for 30 minutes at 4°C. Inoculum was removed, and cells were washed six times with the 4°C PBS solution before RNA extraction. New England Biolabs Universal One-Step RT-qPCR kit was used to prepare cDNA from total RNA. MAYV genomes in the total RNA were quantified through SYBR green-based RT-qPCR on a Bio-Rad CFX96 Touch Real-Time PCR Detection System using the following primers: MAYV 5028For. (5' CCTCTGTTAGTCCTGT GCAATAC 3'); MAYV 5108Rev. (5' AAGGTGCTTAGGGAGCTACT 3'); GAPDHFor. (5' CCAGGTGGTCTCCTCTGACTT 3'); GAPDHRev. (5' GTTGCTGTAGCCAAATTCGTTGT 3'); rp49For. (5' AAGAAGCGGACGAAGAAGT 3'); rp49Rev. (5' CCGTAACCGATGTTT GGC 3'). The housekeeping primers for GAPDH and rp49 were used with MRC-5-derived RNA and Aag2-derived RNA, respectively. Relative MAYV genomes were calculated by nor-malizing the Ct values of MAYV to the inoculum and the housekeeping gene.

## Binding assays in WT 3T3 and 3T3 Mxra8 KO cells

WT 3T3 and 3T3 Mxra8 KO cells at near confluency were pre-chilled at 4°C for 30 minutes. Cells were washed on ice twice with cold PBS containing 5% w/v BSA then inoculated with either WT MAYV or MAYV E2-T179N diluted in PBS with 5% w/v BSA at a MOI of 0.1.

Adsorption occurred at 4°C for 30 minutes. After adsorption, the inoculum was removed, and cells were washed six times with cold PBS with 5% w/v BSA. To remove cells, a solution of 0.1% BSA and 0.3% IGEPAL CA-630 was added to the cells. After 5 minutes on ice, the cells were taken off the wells by pipetting. To analyze virus bound to the cells, viral genomes were quantified using a SYBR green-based RT-qPCR assay on an Applied Biosystems QuantStudio 3 using the following primers: MAYV 5028For. (5' CCTCTGTTAGTCCTGTGCAATAC 3'); MAYV 5108Rev. (5' AAGGTGCTTAGGGAGCTACT 3'); GAPDHFor. (5' CCAGGTGGTCT CCTCTGACTT 3'); GAPDHRev. (5' GTTGCTGTAGCCAAATTCGTTGT 3'). Relative MAYV genomes were calculated by normalizing the Ct values of MAYV to the housekeeping gene.

### Infection in HEK-293A cells exogenously expressing receptors

HEK-293A cells were transfected when at 70% confluency using JetOPTIMUS reagent (Poly-plus) according to the manufacturer's protocols. Expression constructs encoding SLC1A5, hTIM4, hApoER2 isoform 1, hApoER2 isoform 2, hVLDLR, hMxra8, and hACE2 receptor were supplied to the cells at a 1:1 DNA:reagent ratio. Twenty-four hours post-transfection, cells were pre-chilled then inoculated with WT MAYV or MAYV E2-T179N at a MOI of 0.1 while on ice. The cells were placed at 4°C for 30 minutes to synchronize binding. The inoculum was removed and cells washed three times with PBS to remove unbound virus. Media was added to the cells, and an initial supernatant sample was collected. The cells were incubated at 37°C for the next 24 hours post-infection before collecting another supernatant sample. Infectious virus in the supernatant was quantified by plaque assay.

### Mouse infection

Four-week-old, female CD-1 mice were inoculated with $10^5$ PFU/mL of either WT MAYV or MAYV E2-T179N in 50 μL of RPMI-1640 via injection to the left, hind footpad; in the same manner, a mock-infected group was injected with RPMI-1640. At days 1–3 p.i. blood was collected via submandibular bleed and the serum titrated for viremia. Weights of the mice were taken and footpad widths were measured using digital calipers daily following infection. At seven d.p.i. a subset of mice was euthanized for collection of the left, hind footpad for histopathology. Footpad tissue was fixed in 4% formalin solution, pH 6.2 before decalcification in 10% EDTA pH, 7.3 at 4°C. Tissues were hematoxylin- and eosin-stained, imaged and assigned scores by a board-certified pathologist according to the extent of myofiber loss. The remaining mice were euthanized at 9 d.p.i., and footpads were collected and processed as mentioned above.

### Mosquito rearing

Laboratory colonies of *Ae. aegypti* were established from field collections in Kamphaeng Phet Province, Thailand or from a colony isolated from Guerrero, Mexico. All the experiments were performed within 20 generations of laboratory colonization. The insectary conditions for Kamphaeng Phet colony maintenance were 28°C, 70% relative humidity, and a 12h light:12h dark cycle. For Guerrero colonies, maintenance was 28°C, 70–80% relative humidity, and a 12h light:12h dark cycle. All adults were maintained with permanent access to a 10% sucrose solution. Kamphaeng Phet adult females were offered commercial rabbit blood (BCL) twice a week through a membrane feeding system (Hemotek).

### *Ae. aegypti* Kamphaeng Phet population infections

6–8 days old Kamphaeng Phet female mosquitoes were selected and starved overnight prior to the bloodmeal. They were fed with an infectious bloodmeal consisting of 2 mL of previously

washed human blood (ICareB platform, Institut Pasteur), 1 mL of MAYV viral solution ($3x10^6$ PFU diluted in L15 media with P/S, NEAA, 10% FBS and 1% sodium bicarbonate) and 5 mM ATP (Sigma). Mosquitoes were offered the infectious bloodmeal for 20 min through a membrane feeding system (Hemotek) set at 37˚C with a piece of hog gut as the membrane. Following the bloodmeal, fully engorged females were counted, selected, and kept at 28˚C in 70% relative humidity and under a 12 h light: 12 h dark cycle with permanent access to 10% sucrose. 7 days after the bloodmeal, mosquitoes were cold-anesthetized and dissected. Legs and wings were collected on ice. Mosquitoes were salivated for 20 min in a tip containing 20 µL of FBS (Gibco), and midguts were collected. Body parts were collected in microtubes (Qiagen) containing 5 mm diameter stainless steel beads (Qiagen) and 150 µL DMEM supplemented with 2% FBS and homogenized using a TissueLyser II (2 cycles at 30 Hz, 1 min, Qiagen). Homogenates were clarified by centrifugation and frozen until plaque assay titration. Viral loads in the saliva were further amplified on BHK-21 or C6/36 for 3 days, and presence of the virus was determined by direct visualization of cytopathic effects or RT-qPCR detection in the conditions described above for competition assays.

### *Ae. aegypti* Guerrero population infections

3–5 days old Guerrero [53] female mosquitoes were cold anesthetized and sorted 2 days prior to infection. Mosquitoes were starved for 24 hours then fed with an infectious bloodmeal of defibrinated sheep blood consisting of 0.05 mM ATP and $10^7$ PFU/mL of either WT or MAYV E2-T179N diluted in viral diluent. The mosquitoes were fed for 30 minutes through a membrane feeder supplying the infectious bloodmeal at 37˚C with a piece of hog gut. Fully engorged mosquitoes were sorted and maintained in an environmental chamber set to 26˚C, 70–80% relative humidity, and a 12hr light:12hr dark cycle. Mosquitoes were anesthetized at 10 d.p.i., and bodies, legs and wings, and saliva were collected. Bodies and legs and wings were collected in tubes containing a metal bead and RPMI-1640 with 2% FBS, 10 mM HEPES, Amphotericin B (2.5 µg/mL), and gentamicin sulfate (50 µg/mL) (mosquito diluent). Live salivation occurred for one hour in Type A immersion oil which was collected in mosquito diluent and frozen. Tissues were homogenized using a TissueLyser II (30 Hz/s, 2 min, Qiagen). Homogenates were clarified by centrifugation and frozen until plaque assay titration or RT-qPCR on all samples.

### *In silico* N-glycosylation predictions

N-glycosylation prediction was performed using two webservers: NetNGlyc 1.0 (http://www.cbs.dtu.dk/services/NetNGlyc/) and NGlycPred (https://bioinformatics.niaid.nih.gov/nglycpred/). For both servers, we used the MAYV E2 glycoprotein sequence from strain IQT4235. With the NetNGlyc server, we used the default option to run predictions only on the Asn-Xaa-Ser/Thr sequons and used a 0.5 threshold. In the NGlycPred server, we used the preferred option to consider structural properties and patterns.

### Molecular dynamics simulations and calculation of binding free energy values

All the structural models of MAYV in complex with the human Mxra8 receptor were built based on the coordinates position of CHIKV crystallographic structure in complex with Mxra8 (PDB ID: 6JO8 [47]). To model the complexes between MAYV E2 subdomain and Mxra8 D2 domain, we removed from 7KO8 and 6JO8 structures all residues not included in those domains. The asymmetric unit of MAYV Cryo-EM structure (PDB ID: 7KO8) has 4 E1-E2 heterodimer copies whereas CHIKV crystallographic structure (PDB ID: 6JO8) has three copies. Thus, we cross-aligned, one by one, the E2 B subdomains of MAYV structure to

each E2 B subdomains of CHIKV with the Chimera software [75]. This cross-alignment returned 12 complexes between MAYV E2 B subdomains and Mxra8 D2 domain. We estimated predicted free energy changes upon T179N mutation for all 12 MAYV complexes using FoldX 5 software [76]. Before energy calculations, each structure was treated to optimize side-chain rotamers using the FoldX function *RepairPDB*. In FoldX, wild-type structures were mutated and the ΔΔG (in kcal/mol) was estimated by subtracting the resulting energy of the mutated complex by the wild-type complex. We reported the FoldX result as mean ± standard deviation of ΔΔG for the 12 MAYV complexes. These same WT and T179N complexes were used as initial structures for molecular dynamics simulations using Amber 20 suit of programs with the *ff14SB* force field. Using *tleap*, we added Na+ and Cl- counter ions to reach net-neutralization with salt excess to reach 150 mM NaCl [77]. Each structure was solvated in a truncated octahedral box (15 A from the solute) filled with *TIP3P* water. The *PBRadii* was set to *mbondi2* with *tleap* program. The system was minimized by 2500 steps of steepest descent minimization and 2500 steps of conjugate gradient. The equilibration was performed using the NVT ensemble (200 ps) followed by NPT ensemble (200 ps), both with harmonic restraints in protein atoms. A last NPT equilibration step without restraints was performed for 500 ps. For each complex, the production run was performed at 298 K for 2 ns with a time step of 0.002 fs in triplicate. Hydrogen-containing bonds were constrained using SHAKE [78]. Long-range electrostatic interactions were calculated using particle Mesh Ewald and short-range non-bonded interactions were calculated with a 9 A cutoff. In all simulation steps we applied a harmonic restraint of 10 kcal·mol−1 ·Å −2 to the backbone atoms to prevent an overall displacement of the complex from its initial position. We estimated the relative binding free energy using 1-trajectory Molecular Mechanics Generalized Born Surface Area (MM/GBSA) calculations. For this, we used 20 frames from entire 2 ns simulations with MMPBSA.py script [79]. The individual topology files for the calculations were generated using ante-MMPBSA.py script. Since our objective was only to compare wild-type and mutant interactions, we did not include the entropy term. As a result, the binding free energy estimate reported a relative value, not an absolute. In MMGBSA calculations, we used igb model 5 (all the other parameters were default). We reported the MMGBSA results as a ΔΔG of the difference between the relative ΔG estimated for T179N MAYV:Mxra8 and WT MAYV:Mxra8. To model the entire MAYV spike, formed by three E1-E2 heterodimers (not including transmembrane domains), in complex with Mxra8, we aligned individually the MAYV E1-E2 heterodimers (from PDB ID: 7KO8) with CHIKV E1-E2 heterodimers (from PDB ID: 6JO8) in Chimera. Then, we created the MAYV T179N mutated structure with the YASARA software [80]. Both wild-type and mutant MAYV trimeric complexes were submitted to energy minimization in solvent with YASARA to obtain an initial model. Then, we performed a 20 ns molecular dynamics simulation in quintuplicate for WT and T179N MAYV complexes, following the same simulation protocol described above without backbone restraints. The MMGBSA calculation was performed using 100 frames from the last 4 ns with the same parameters described above. Before any calculations or simulations, the protonation state at pH 7.4 for each residue of the structural models was assigned using the *pdb2pqr30* script [81] and *propka* as the titration state method [82]. For all cases, we also modeled the missing loop of Mxra8 D2 domain using YASARA software [80]. The convergence of the simulations was accessed by the RMSD of the Mxra8 backbone in a trajectory aligned by the MAYV backbone (**S7E and S7F Fig**).

## Statistical analysis

All data were analyzed using the Prism 9 software (GraphPad) and are presented as mean ± standard deviation unless indicated otherwise. The statistical tests used are described

in each figure legend and were performed using GraphPad Prism. All experiments (except serial and DMS passaging) were performed in two to three independent replicates with at least three technical replicates per group.

## Supporting information

**S1 Fig. Natural evolution of Mayaro virus (MAYV) in *Anopheles gambiae* cells.** Experimental evolution was performed using traditional serial passaging in 4a-3A cells. We serially passaged MAYV at a MOI of 0.01 for a total of ten passages. Following one, five, and ten passages, the viral RNA was sequenced using Illumina NGS to identify potentially adaptive mutations. No high frequency variants were identified following one passage, and as such are not depicted here.
(TIF)

**S2 Fig. MAYV deep mutational scanning (DMS) populations are highly diverse and replicate *in vitro* and *in vivo*. A.** Genome organization of MAYV DMS viruses. Created with BioRender.com. **B.** Non-synonymous nucleotide diversity for wild-type (WT) and MAYV DMS. The three independent MAYV DMS populations were combined to aid visualization. **C-D.** Three passages were performed *in vitro* at an MOI of 0.01 in LLC-MK2 (monkey kidney; **C**) and Aag2 (*Aedes aegypti* mosquito; **D**). **E.** Viremia of WT MAYV and DMS populations in mice during passaging. **F.** Viral titers in saliva of WT MAYV and DMS populations after passage in *Ae. aegypti* mosquitoes. Statistical analysis: * = $p < 0.05$; ** = $p < 0.01$; **** = $p < 0.0001$ (one-way ANOVA with Dunnett's correction).
(TIF)

**S3 Fig. Deep mutational scanning (DMS) of Mayaro virus (MAYV) in live mice and mosquitoes. A-B.** The three MAYV DMS populations, along with WT MAYV, were used to perform three passages in mice (**A**) and *Ae. aegypti* mosquitoes (**B**). Following passage, the viral RNA was sequenced, and selection analyses were performed to identify enriched variants. The top three variants based on selection strength, are presented for mice (**A**). Given the low level of enrichment observed in *Ae. aegypti* mosquitoes, we do not present individual variants.
(TIF)

**S4 Fig. Variants enriched in live mice and mosquitoes have little effect on viral fitness. A.** List of the viruses used for the competition assays. All mutations were identified from deep mutational scanning data, and the phenotypes observed for the different viruses are indicated. The wild-type data—included as a comparison—is the same as is presented in Fig 2 in the main text. **B-E.** Competition assays in Aag2 (**B**), U4.4 (**C**), LLC-MK2 (**D**), and MRC-5 (**E**) cells. Cells were infected at a MOI of 0.01 using a 1:1 ratio based on PFUs for each mutant or WT with a genetically marked MAYV competitor virus. Viral supernatants were harvested at 72h post-infection for Aag2 and 48h post-infection for the other cell lines. Replication of WT and mutant viruses was assessed by RT-qPCR using specific probes labeled with different fluorophores. $Log_{10}$ fitness was calculated by normalizing replication of each virus against a genetically marked reference virus. The mean of 4 independent experiments is represented with standard deviation. Statistical analysis: * = $p < 0.05$; ** = $p < 0.01$; **** = $p < 0.0001$ (one-way ANOVA with Dunnett's correction). Consensus change in some mosquito saliva means that the mutation was found at above 50% in one mosquito saliva sample: E2 L414I was found at 89%, E3-P34T was found at 99.7%, and E3-K65A was found at 99.7%.
(TIF)

**S5 Fig. WT MAYV and MAYV E2-T179N have similar genome:PFU ratios following Aag2 infection.** Aag2 cells were infected with either WT MAYV or MAYV E2-T179N at a MOI of

0.1, and genome:PFU ratios were measured each day post infection via RT-qPCR and plaque assay. The -1 day post infection represents the genome:PFU ratio of the inoculum used for the infection. Genome:PFU ratios represented are $\log_{10}$-transformed. The means of two independent experiments are represented, and error bars represent standard deviation. Statistical analysis: non-significant (two-way ANOVA with Šídák's multiple comparisons test).
(TIF)

**S6 Fig. Binding affinity of WT and MAYV E2-T179N MAYV to Aag2 cells.** Binding assays of WT MAYV and MAYV E2-T179N were performed by inoculating chilled Aag2 cells with virus at a MOI of 0.1 prior to adsorption at 4˚C. Unbound virus was washed away, and bound virus was quantified by RT-qPCR using RNA extracted from the inoculated cells. The relative quantities of MAYV genomes were determined by normalizing to the Ct values of a housekeeping gene and the Ct value of virus in the inoculum. Data represent the means of two independent experiments. Y-axis is log-transformed for clearer visualization. Statistical analysis: *** = $p < 0.002$ (unpaired t-test).
(TIF)

**S7 Fig. Structural analysis of WT MAYV and MAYV E2-T179N Spike and complexation with Mxra8. A.** MAYV spike structure (PDB ID: 7KO8) composed of three E1-E2 heterodimers. Only the E1-E2 ectodomains are presented. For clarity, we showed only one heterodimer (E1 in green and E2 in magenta) in cartoon representation, and the other two are represented as a surface. Zoomed-in views of T179 and the T179N mutation are shown in insets. T179, T179N and the E1 protein are represented in spheres. **B**. MAYV spike in complex with three copies of the human Mxra8 receptor (in yellow) obtained from the last frame of the 20 ns molecular dynamics simulations. For clarity, only one E1-E2 heterodimer is represented as a cartoon. Inset shows a side view of the interface between the E2 B subdomain and the Mxra8 receptor. T179 is shown in spheres. All images were generated using the ChimeraX software. **C.** Binding free energy changes (ΔΔG in kcal/mol) of 12 MAYV E2 B:Mxra8 D2 subdomain complexes (n = 12) upon MAYV T179N mutation using FoldX and MM/GBSA methods. ΔΔG was obtained by subtracting the ΔG estimated for T179N MAYV:Mxra8 interaction from the ΔG obtained for WT MAYV:Mxra8 interaction. MMGBSA calculations were performed from 2 ns molecular dynamics simulation (in triplicate) of each complex. In none of the applied methods did the estimated ΔΔG significantly differ from zero (p > 0.05—one sample t-test). The box plot central line indicates the median and the red circle indicates the mean. **D**. Binding free energy changes (ΔΔG in kcal/mol) of the full trimeric MAYV spike in complex with the full Mxra8 ectodomain upon MAYV T179N mutations using MM/GBSA method. The calculations were performed from a quintuplicate of 20 ns molecular dynamics simulations (n = 5) (see methods for details). ΔΔG was obtained by subtracting the relative ΔG estimated for T179N MAYV:Mxra8 interaction from the ΔG obtained for WT MAYV:Mxra8 interaction. The estimated ΔG did not significantly differ from zero (p > 0.05—one sample t-test). The box plot central line indicates the median, and the red circle indicates the mean. **E-F.** Root-mean-square deviation (RMSD) of Mxra8 receptor in complex with (**E**) WT MAYV spike or (**F**) MAYV T179N during 20 ns molecular dynamics simulations. To calculate the RMSD of Mxra8 backbone atoms, the trajectory was aligned by the MAYV backbone atoms using the first frame as reference in VMD. Each line corresponds to a replicate, and the vertical dashed line indicates the last 4 ns of the simulation.
(TIF)

**S1 File. Envelope variants identified from serial passaging.**
(XLSX)

**S2 File. Log2 selection of envelope variants identified from passaging DMS populations by DMS site and host species.**
(XLSX)

**S3 File. Log2 selection of envelope variants identified from passaging DMS populations by variant.**
(XLSX)

**S4 File. Mutagenic primer sequences for generating DMS populations.**
(XLSX)

## Acknowledgments

The authors thank Rodrigo Guabiraba for helpful discussions. We are grateful to Jonathan Abraham of Harvard Medical School and Hyeryun Choe of the The Scripps Research Institute for sharing expression plasmids.

### Disclaimer

The content of the information does not necessarily reflect the position or the policy of the U. S. government, and no official endorsement should be inferred.

## Author Contributions

**Conceptualization:** Ferdinand Roesch, Lucía Carrau, Maria-Carla Saleh, Marco Vignuzzi, James Weger-Lucarelli.

**Data curation:** Ferdinand Roesch, Cassandra Koh, Jesse D. Bloom, James Weger-Lucarelli.

**Formal analysis:** Chelsea Cereghino, Ferdinand Roesch, Lucía Carrau, Helder V. Ribeiro-Filho, Jeffrey M. Marano, Anne M. Brown, Tanya LeRoith, Jesse D. Bloom, Rafael Elias Marques, James Weger-Lucarelli.

**Funding acquisition:** Maria-Carla Saleh, Marco Vignuzzi, James Weger-Lucarelli.

**Investigation:** Chelsea Cereghino, Ferdinand Roesch, Lucía Carrau, Alexandra Hardy, Helder V. Ribeiro-Filho, Annabelle Henrion-Lacritick, Cassandra Koh, Tyler A. Bates, Pallavi Rai, Christina Chuong, Shamima Akter, Thomas Vallet, Hervé Blanc, Truitt J. Elliott, Anne M. Brown, Tanya LeRoith, Jesse D. Bloom, James Weger-Lucarelli.

**Methodology:** Chelsea Cereghino, Ferdinand Roesch, Lucía Carrau, Alexandra Hardy, Helder V. Ribeiro-Filho, Cassandra Koh, Anne M. Brown, Pawel Michalak, Jesse D. Bloom, Rafael Elias Marques, Maria-Carla Saleh, Marco Vignuzzi, James Weger-Lucarelli.

**Project administration:** Marco Vignuzzi, James Weger-Lucarelli.

**Resources:** Maria-Carla Saleh, James Weger-Lucarelli.

**Supervision:** Maria-Carla Saleh, Marco Vignuzzi.

**Validation:** Chelsea Cereghino, Ferdinand Roesch, James Weger-Lucarelli.

**Visualization:** Helder V. Ribeiro-Filho.

**Writing – original draft:** Chelsea Cereghino, Ferdinand Roesch, Helder V. Ribeiro-Filho, James Weger-Lucarelli.

**Writing – review & editing:** Chelsea Cereghino, Ferdinand Roesch, Lucía Carrau, Helder V. Ribeiro-Filho, Cassandra Koh, Christina Chuong, Anne M. Brown, Pawel Michalak, Jesse

D. Bloom, Rafael Elias Marques, Maria-Carla Saleh, Marco Vignuzzi, James Weger-Lucarelli.

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
