## [Decision Letter · Decision Letter 0]

17 May 2022

Dear Dr. Weger-Lucarelli,

Thank you very much for submitting your manuscript "The E2 glycoprotein holds key residues for Mayaro virus adaptation to the urban Aedes aegypti mosquito" for consideration at PLOS Pathogens. As with all papers reviewed by the journal, your manuscript was reviewed by members of the editorial board and by several independent reviewers. In light of the reviews (below this email), we would like to invite the resubmission of a significantly-revised version that takes into account the reviewers' comments.

The manuscript is being returned with three reviews. While each of the reviewer's comments should be addressed, particular attention should be focused on the following issues:

(1) Both R1 and R2 raised concerns about the virus binding and replication data, including the role of Mxra8, other entry receptors, and the rigor of the experiments (e.g., the need for appropriate positive and negative controls). The authors should provide additional data to explain differences in binding and replication between WT and E2 T179N viruses.

(2) Address the concerns of R3 regarding statements of MAYV emergence and the emergence potential of MAYV E2 T179N, as well as other speculative components of the manuscript.

(3) Issues with reproducibility in mosquito transmission data should be addressed. 

We cannot make any decision about publication until we have seen the revised manuscript and your response to the reviewers' comments. Your revised manuscript is also likely to be sent to reviewers for further evaluation.

Sincerely,

Thomas E. Morrison

Opinions Editor

PLOS Pathogens

Mark Heise

Section Editor

PLOS Pathogens

Kasturi Haldar

Editor-in-Chief

PLOS Pathogens

orcid.org/0000-0001-5065-158X

Michael Malim

Editor-in-Chief

PLOS Pathogens

orcid.org/0000-0002-7699-2064

Reviewer's Responses to Questions

**Part I - Summary**

Reviewer #1: The authors used two different approaches (serial passaging and direct mutational scanning) to identify E2 protein T179N as a key mutation in conferring enhanced replication and fitness of Mayaro virus (MAYV) in Aag2 insect cells but not in human fibroblast cells (MRC-5). The mutation to T179N also occurred during passaging in A. albopictus (U4.4) cells. Structural studies did not reveal any significant differences in binding of E2 protein with and without T179N mutation to Mxra8 receptor. It was concluded that the decrease of MAYV T179N fitness in human cells is not mediated by Mxra8. However, knock-out of Mxra8 in 3T3 cells resulted in decreased replication of MAYV T179N virus. In contrast to MAYV, introduction of the T179N mutation in CHIKV E2 protein resulted in increased replication in both human and insect cells. The mutation to T179N also resulted in attenuation of MAYV in mice infections. The mutation confers advantage only in insect cells and does not result in increased transmission in human population.

Weakness: Although the mutation confers increased fitness in mosquito cells, the mechanisms for increased replication of MAYV T179N in Aag2 cells are not clear.

1. It is mentioned that few mutations were found in protein coding regions during serial passaging. Include information on the coding regions where the mutations were found during serial passaging. Some of the mutations may be important in enhancing replication in Aag2 mosquitoes (Figs 2 and 3).

2. Binding and replication studies - although MAYV E179N binds less to both Aag2 and MRC-5 cells (Fig 4 and S6), replication is enhanced in Aag2 cells (Fig 3). Binding differences of WT and MAYV T179N to MRC-5 cells (from 10000 copies to approx 1000 copies) is more pronounced compared to binding to Aag2 cells (700 copies to 250 copies) which provides an explanation for the decreased replication in MRC-5 cells. The authors will need to explain the decreased binding and increased replication of MAYV T179N to Aag2 cells.

3. Include data on the fitness of MAYV T179N mutant virus derived from serial passaging in U4.4 cells. In comparison to replication in Aag2 cells, include data on the binding and replication of MAYV WT and MAYV T179N mutant virus in U4.4 cells.

4. Fig 4F - replication of CHIKV T179N virus was enhanced in both MRC-5 and Aag2 cells. In comparison to MAYV include data on the replication of CHIKV WT and T179N mutant in Mxra8 KO fibroblasts. Also in Fig 4A include data on the binding of WT CHIKV and CHIKV T179N mutant to Aag2 and MRC-5 cells as a comparison to MAYV. These studies will reveal the presence of receptors other than Mxra8 in alphavirus infections.

5. The authors will need to explore importance of other receptors such as NRAMP2, VLDLR (Rose et al., Cell Host Microbe 2011, 10(2)97-104; Clark et al., Nature 2022, 602; 475–480) in binding of MAYV to MRC-5 and Aag2 cells.

Reviewer #2: The manuscript PPATHOGENS-D-22-00591,The E2 glycoprotein holds key residues for Mayaro virus adaptation to the urban Aedes aegypti mosquito by Roesch et al. describes experimental evolution of MAYV using serial in vitro passaging and ‘facilitated’ evolution via deep mutational scanning (DMS). An E2 gene mutation that increased in frequency in after serial mosquito cell passage was then expressed in an infectious clone and compared to WT via parallel vector competence studies in Aedes aegypti, where it increased transmissibility. Insertion of the same mutation into another alphavirus, chikungunya virus, also increased replication in mosquito cells. In mice, compared to WT MAYV, the mutation reduced viremia and delayed footpad swelling. These findings are interpreted to indicate that adaptation for increased fitness in a vector may come at the cost of reduced fitness in a vertebrate host, which is a conundrum arboviruses must deal with to survive and an impediment to their successful emergence, where mutations that are more fit in both host types are those that cause outbreaks. Understanding viral genetic determinants of MAYV emergence is important and timely given that MAYV has not produced widespread epidemics like CHIKV, and also because MAYV is relatively poorly studied in general. The conclusion-that arboviruses are subjected to fitness costs as they alternate hosts-is not novel, nor is the observation that an envelope gene mutation in an alphavirus mediates a change in fitness. However, this is the first study of this type for MAYV. As such, this manuscript provides new knowledge.

Reviewer #3: In “The E2 glycoprotein holds key residues for Mayaro virus adaptation…” the authors have completed a series of preliminary experiments to try to identify mutations that might arise – given the right circumstances – and cause outbreaks. The work is straightforward and similar to other studies performed by this group and others showing functionally the same outcome – that repeated passage in a single cell type leads to adaptation in that system at the expense of growth in an alternate system. The methods are sound and well established. The writing is clear and fluid.

Interestingly, the one mutation they fixed on (which is unfortunate that they didn’t explore a few of the others identified as well) – actually suggests that it would NOT cause large outbreaks as it actually was attenuated in the vertebrate cells. The authors nicely described this limiting process in the discussion where they spoke about the need for balance in a two-host system to maintain some level of infectivity in each and how adaptation to one host likely reduces the infectivity in the other system – thus, limiting its emergence potential. While the authors do address this point in the discussion, they need to remove all the earlier statements saying this mutant is a likely a driver of emergence or outbreaks as their own work really suggests it is not (several places in the intro and results).

**Part II – Major Issues: Key Experiments Required for Acceptance**

Reviewer #1: 1. Serial passaging resulted in E2 protein T179N mutation in Aag2 and U4.4 cells (A. albopictus). In the fitness studies MAYV T179N enriched from Aag2 cells showed a slight increase in fitness in U4.4 cells (Fig 2C). In this context, the authors must show the fitness of MAYV T179N mutant virus derived from serial passaging in U4.4 cells. In comparison to replication in Aag2 cells, include data on the binding and replication of MAYV WT and MAYV T179N mutant virus in U4.4 cells. Fitness studies indicate that mutation may facilitate the transmission of MAYV by A. albopictus mosquitoes.

2. In insect cells, Mxra8 is not a receptor for alphaviruses. In the absence of any structural differences it is intriguing why there was decreased binding of MAYV T179N to MRC-5 cells. The authors will need to explore importance of other receptors such as NRAMP2, VLDLR (Rose et al., Cell Host Microbe 2011, 10(2)97-104; Clark et al., Nature 2022, 602; 475–480) in binding of MAYV to MRC-5 and Aag2 cells. NRAMP2 is required for Sindbis virus binding, entry, and infection in both mammalian and insect cells. Caluwe et al., Front Microbiol 2021, 12: Art 615165 have reported that CD147 complex proteins are involved in the entry of CHIKV and MAYV.

3. Structural studies did not reveal any differences between E2 T179N and WT E2 protein binding to Mxra8 (Fig 4C,D and E). However, replication of MAYV T179N was reduced in Mxra8 3T3 KO cells compared to 3T3 cells infected with WT virus, which indicate the importance of Mxra8 in MAYV replication. Data showing the binding of MAYV T179N to WT and Mxra8 KO 3T3 cells must be included.

Reviewer #2: 1. While most of the conclusions are supported by the data, there are several exceptions where this is not the case. Notably:

a. The statement: “However, using this model, the absence of Mxra8 induced a similar reduction of viral replication of both WT and E2-T179N MAYV, confirming our in silico analyses showing that the reduced viral binding of E2-T179N MAYV is Mxra8-independent” is not supported by the data shown in Fig 4F. MAYV E2-T179N 3T3 Mrxra8 KO titers appear lower at most/all time points compared to MAYV WT 3T3 Mrxra8 KO. The legend states that statistical tests showed no differences, but this is hard to believe given that the error bars for the MAYV E2-T179N 3T3 Mrxra8 KO titers do not overlap with any of the other groups on days 1, 2, 3 and 4. Wouldn’t the data shown suggest that binding of MAYV E2 T179N is in fact at least partially Mxra8 dependent? Also missing are positive and negative controls for this experiment.

b. Scientific rigor is lacking for Fig 5 which does not show Ns for each of the three experiments, which could help the reader evaluate whether the higher transmission rate of 60% in one MAYV E2-T179N replicate is an artifact of smaller cohort size and therefore more prone to stochastic effects, especially since transmission rates in the other 2 replicates appear much lower.

c. Since mice inoculated with MAYV E2-T179N had a nearly identical pattern of footpad swelling to WT that was delayed by 1 day, the conclusion “These results are consistent with E2-T179N causing an attenuation of MAYV in the mammalian host” is not entirely supported. Delayed, perhaps, but clinical disease in the MAYV E2-T179N mice as assessed by swelling does not seems quantitatively different from WT, which doesn’t support use of ‘attenuation’.

d. For full transparency, the scores of mice selected as ‘representative’ for images 1 and 2 in Fig 6E should be shown.

2. Use of the words ‘mosquito’ and ‘mouse’ in several places, while data are being described from mosquito cells and mouse cells, is misrepresentative of the systems in which the experimental evolution/DMS studies were performed. For example:

a. Fig 2: Clarify that Aag2 and U4.4 are mosquito cell lines and not mosquitoes (as text in parentheses seems to indicate).

b. Fig S4: Figure heading; clarify that it was mosquito and mouse cells, not entire organisms.

c. P13: ‘We also evaluated the fitness of mutations enriched following passage through mice and mosquitoes and observed’…should be mouse and mosquito cells.

3. The Mxra8 data seems a bit of a distraction from the central message of the manuscript in that: 1) it does not explain the less efficient binding of MAYV E2-T179N mutant to human cells, 2) the same phenotype was observed in mosquito cells, and 3) it is unknown known (although probably likely) whether MAYV uses Mxra8 as a primary entry receptor.

Reviewer #3: A few other key points of concern (for items throughout the manuscript) are noted below for the authors to consider.

1) In the introduction, the authors tend to overstate the risk of MAYV emergence. As these are all speculative statements, it is suggested that the authors temper these statements with modifiers such as “may” lead to emergence, or “limited MAYV circulation”, etc. In several places, it already sounds like a huge problem; there is no need to try to oversell this preliminary work.

2) While some level of speculation is ok in the discussion, the authors really should consider removing most of their “speculations” throughout the results section. For each section, they report their results and then say “this is likely due to…” or “this likely leads to…” (or similar sentiments). Since no work was performed to elucidate mechanisms, it is not really appropriate to add such wild speculation without any supporting data. Provide the data you obtained, state more work is needed to understand the mechanisms, and leave it at that. This has something of the feel of a dissertation rather than a scientific publication.

3) While perhaps slightly eluded to in the discussion, the authors really fail to mention the role of the remaining viral “backbone” in contributing to the effects of any specific point mutations. Thus, STRAIN is incredibly important and the authors need to mention this. As an example, they refer to CHIKV mutation that leads to altered vector species competence yet they failed to state that this same mutation had no effect in the context of a CHIKV backbone from a different genotype. Asian genotype strains were not affected in the same way as ECSA genotype strains by the same mutations so introduction of this single mutation into any particular clone may not result in a generalizable outcome. This needs to be addressed in the discussion as well particularly since the DMS work focused only on the glycoprotein sections of the genome.

4) There are a few places in the results where the authors seem to discount “contrary” results and focus only on those data that support their speculations. One example is discounting the finding that “peak footpad swelling was similar between the groups”. This seems to later be ignored in the speculative part of this section. Similarly, it is concerning that there is so little reproducibility in mosquito transmission data (15-60% is huge variation!) yet this concern is not addressed. If the 15% set is considered, this is not significantly different from the control findings. The authors need to use caution in failing to include all the results in their interpretations. Let the data speak for itself and include it all in your discussion.

**Part III – Minor Issues: Editorial and Data Presentation Modifications**

Reviewer #1: 1. Fig S4 – please clarify the meaning of some mouse serum and mouse saliva?

2. The E2 mutation T179N resulted in attenuation in MAYV infections in mice. It will be interesting to examine whether the mutation also results in attenuation of CHIKV infections in mice.

Reviewer #2: 4. The introduction and results sections lack details on how the DMS works and how it was performed. This makes it difficult for a reader unfamiliar with references 36-38 to understand this approach. Even a brief background/rationale would be helpful.

5. The rationale for use of Anopheles cells would enable the reader to understand whether the idea was to determine whether MAYV could adapt to multiple mosquito genera (i.e. ref. 35?), or whether this was intended as a negative control, especially since the data from those cells are sort of sidelined by relegation to Figure S1 but nowhere else.

6. Fig S3B: It is unclear which dots show the amino acids indicated.

7. P8 ‘suggesting a mammalian-specific attenuation’; a bit of a stretch given just 2 cell types were evaluated, suggest tempering language.

8. Replacing the labels ‘1’ with ‘WT’ and ‘2’ with ‘E2-T179N’ would make Fig6E more readable.

Reviewer #3: (No Response)

PLOS authors have the option to publish the peer review history of their article (what does this mean?). If published, this will include your full peer review and any attached files.

Reviewer #1: No

Reviewer #2: No

Reviewer #3: No
---

## [Decision Letter · Decision Letter 1]

1 Mar 2023

Dear Dr. Weger-Lucarelli,

Thank you very much for submitting your manuscript "The E2 glycoprotein holds key residues for Mayaro virus adaptation to the urban Aedes aegypti mosquito" for consideration at PLOS Pathogens. As with all papers reviewed by the journal, your manuscript was reviewed by members of the editorial board and by several independent reviewers. The reviewers appreciated the attention to an important topic. Based on the reviews, we are likely to accept this manuscript for publication, providing that you modify the manuscript according to the review recommendations.

Based on the nature of these studies and the findings reported, we request that two issues be addressed:

(1) Please confirm IBC approval for the experiments reported in this manuscript.

(2) Please provide more detailed information about the biosafety practices used during the course of these experiments.

Sincerely,

Thomas E. Morrison

Academic Editor

PLOS Pathogens

Mark Heise

Section Editor

PLOS Pathogens

Kasturi Haldar

Editor-in-Chief

PLOS Pathogens

orcid.org/0000-0001-5065-158X

Michael Malim

Editor-in-Chief

PLOS Pathogens

orcid.org/0000-0002-7699-2064

Based on the nature of these studies and the findings reported, we request that two issues be addressed:

(1) Please confirm IBC approval for the experiments reported in this manuscript.

(2) Please provide more detailed information about the biosafety practices used during the course of these experiments.

Reviewer Comments (if any, and for reference):

Reviewer's Responses to Questions

**Part I - Summary**

Reviewer #1: A majority of the authors' responses to the reviewer's comments are satisfactory. The authors conducted additional experiments in response to the reviewer's comments. The text and figure legends have been updated with appropriate explanations and descriptions.

Reviewer #2: (No Response)

Reviewer #4: Cereghino and colleagues aimed to study how Mayaro virus (MAYV) could potentially exhibit mutations and, therefore be transmitted by the urban Aedes aegypti mosquito. The authors performed a mix of robust experimental evolution and genomic approaches to identify a key mutation in the virus receptor binding (E2) protein. They have worked with different cell lines, mosquito populations as well as mouse model to show that the virus exhibiting the E2 mutation, may result in increased vector competence, a potential step for future vector adaptation and therefore, virus spread.

This is the first time I see this manuscript, as it has been previously reviewed by three other reviewers (R1). After checking the previous revision of three reviewers and going through the last version of the manuscript, which contains thorough improvements on the text as well as the addition of other experiments, I believe the current version is suitable for publication in Plos Pathogens.

**Part II – Major Issues: Key Experiments Required for Acceptance**

Reviewer #1: The authors have examined the importance of other alphavirus receptors in the binding of mutant and WT Mayaro virus.

Reviewer #2: (No Response)

Reviewer #4: n/a

**Part III – Minor Issues: Editorial and Data Presentation Modifications**

Reviewer #1: (No Response)

Reviewer #2: (No Response)

Reviewer #4: n/a

PLOS authors have the option to publish the peer review history of their article (what does this mean?). If published, this will include your full peer review and any attached files.

Reviewer #1: No

Reviewer #2: No

Reviewer #4: No

Figure Files:

Data Requirements:

Reproducibility:

References:

---

## [Editor Report · Decision Letter 2]

13 Mar 2023

Dear Dr. Weger-Lucarelli,

We are pleased to inform you that your manuscript 'The E2 glycoprotein holds key residues for Mayaro virus adaptation to the urban Aedes aegypti mosquito' has been provisionally accepted for publication in PLOS Pathogens.

Best regards,

Thomas E. Morrison

Academic Editor

PLOS Pathogens

Mark Heise

Section Editor

PLOS Pathogens

Kasturi Haldar

Editor-in-Chief

PLOS Pathogens

orcid.org/0000-0001-5065-158X

Michael Malim

Editor-in-Chief

PLOS Pathogens

orcid.org/0000-0002-7699-2064
---

## [Editor Report · Acceptance letter]

31 Mar 2023

Dear Dr. Weger-Lucarelli,

We are delighted to inform you that your manuscript, "The E2 glycoprotein holds key residues for Mayaro virus adaptation to the urban Aedes aegypti mosquito," has been formally accepted for publication in PLOS Pathogens.

Best regards,

Kasturi Haldar

Editor-in-Chief

PLOS Pathogens

orcid.org/0000-0001-5065-158X

Michael Malim

Editor-in-Chief

PLOS Pathogens

orcid.org/0000-0002-7699-2064